

# Assessment of Mixed-Layer Height Estimation from Single-wavelength Ceilometer Profiles

Travis N. Knepp[1,2], James S. Szykman[3,4], Russell Long[3], Rachelle M. Duvall[3], Jonathan Krug[3], Melinda Beaver[3], Kevin Cavender[3], Keith Kronmiller[5], Michael Wheeler[5], Ruben Delgado[6], Raymond Hoff[6], Timothy Berkoff[2], Erik Olson[7], Richard Clark[8], Daniel Wolfe[9], David Van Gilst[10], and Doreen Neil[2]

[1]Science Systems and Applications Inc., Hampton, Virginia 23666, USA
[2]NASA Langley Research Center, Hampton, Virginia 23681, USA
[3]US EPA, Research Triangle Park, Durham, North Carolina 27709, USA
[4]Currently assigned to NASA Langley Research Center, Hampton, Virginia 23681, USA
[5]Jacobs Technology Inc., Tullahoma, Tennessee 37388, USA
[6]Joint Center for Earth Systems Technology, University of Maryland Baltimore County, Baltimore, Maryland 21250, USA
[7]Space Science and Engineering Center, University of Wisconsin-Madison, Madison, Wisconsin 53706, USA
[8]Department of Earth Sciences, Millersville University, Millersville, Pennsylvania 17551, USA
[9]NOAA/ESRL Physical Sciences Division, Boulder, Colorado 80305, USA
[10]National Suborbital Education and Research Center, University of North Dakota, Grand Forks, North Dakota 58202, USA

*Correspondence to:* T. N. Knepp
(travis.n.knepp@nasa.gov)

**Abstract.** Differing boundary/mixed-layer height measurement methods were assessed in moderately-polluted and clean environments, with a focus on the Vaisala CL51 ceilometer. This intercomparison was performed as part of ongoing measurements at the Chemistry And Physics of the Atmospheric Boundary Layer Experiment (CAPABLE) site in Hampton, Virginia and during the 2014 Deriving Information on Surface Conditions from Column and Vertically Resolved Observations Relevant to

5   Air Quality (DISCOVER-AQ) field campaign that took place in and around Denver, Colorado. We analyzed CL51 data that were collected via two different methods (BLView software, which applied correction factors, and simple terminal emulation logging) to determine the impact of data collection methodology. Further, we evaluated the STRucture of the ATmosphere (STRAT) algorithm as an open-source alternative to BLView (note that the current work presents an evaluation of the BLView and STRAT algorithms and does not intend to act as a validation of either). Filtering criteria were defined according to the

10   change in mixed-layer height (MLH) distributions for each instrument and algorithm and were applied throughout the analysis to remove high-frequency fluctuations from the MLH retrievals. Of primary interest was determining how the different data-collection methodologies and algorithms compare to each other and to radiosonde-derived boundary-layer heights when deployed as part of a larger instrument network. We determined that data-collection methodology is not as important as the processing algorithm and that much of the algorithm differences might be driven by local meteorology and precipitation events

15   that pose algorithm difficulties. The results of this study show that a common processing algorithm is necessary for LIght Detection And Ranging (LIDAR)-based MLH intercomparisons, and ceilometer-network operation and that sonde-derived boundary layer heights are higher (10–15% at mid-day) than LIDAR-derived mixed-layer heights. We show that averaging



the retrieved MLH to 1-hour resolution (as necessary for a priori data model initialization) significantly improved correlation between differing instruments and differing algorithms.

# 1 Introduction

The atmospheric boundary layer (ABL) is the lowermost portion of the troposphere that is directly influenced by the Earth's surface and responds to surface forcing of heat, moisture, pollutant emissions, and momentum on a timescale of 1 hour or less (Stull, 1988). The ABL can be defined by a number of criteria depending on the particular interest (e.g. thermodynamic boundary layer, chemical boundary layer (CBL), aerosol mixed layer). The ABL is typically defined by thermodynamic data (i.e., potential temperature and/or skew-T plot) obtained from meteorological sondes. While meteorological sondes have excellent vertical resolution, the temporal resolution is generally poor, ongoing regular sonde launches are labor intensive, and coverage is limited. Conversely, mixed-layer heights (MLH), as calculated from backscatter LIght Detection And Ranging (LIDAR) instruments, provide both excellent vertical and temporal resolution. Typical analysis of LIDAR data involves identification of gradients within the aerosol profile (Brooks, 2003), which is generally considered to be a marker for the MLH. With respect to air quality, the top of the ABL often acts like a lid on the lowest layer of the atmosphere and temporarily traps the majority of near-surface anthropogenic and biogenic emissions. As a result, the vertical distribution of ambient air pollutants, and associated precursors, within the ABL and lower-troposphere are strongly influenced by the height of, and vertical mixing within, the ABL.

ABL variability complicates quantitative determination of surface trace-gas levels from a remote-sensing platform (Coen et al., 2014; Herman et al., 2009; Knepp et al., 2015; Lamsal et al., 2008, 2014; Petritoli et al., 2004; Piters et al., 2012). Therefore, properly accounting for ABL variability from a continuous measurement system such as LIDAR will provide invaluable information to policy, health, modeling, and remote-sensing communities for applications sensitive to the vertical profiles of tracers (Compton et al., 2013; Martin, 2008; Scarino et al., 2014). In 2009, the United States National Research Council highlighted ABL height as a high priority observation needed to improve meso-scale predictions of air quality, short-range severe-weather forecasting, and regional climate modeling. More recently, the National Plan for Civil Earth Observation called for improved observation density and sampling of the boundary layer (Science and , NSTC). In 2015, as part of the revisions to the ozone ($O_3$) National Ambient Air Quality Standards, the U.S. Environmental Protection Agency (EPA) finalized a new requirement under the Photochemical Assessment Monitoring Stations (PAMS) program for the collection of continuous MLH observations. By 2019, the PAMS program will involve implementation of approximately fifty air-quality sites in the United States that provide continuous MLH.

Kotthaus et al. (2016) have shown that intercomparison of ceilometer data is not a straight-forward endeavor. In preparation for upcoming PAMS changes, an intercomparison of ceilometer instrumentation was carried out. Presented here are results from an intercomparison of three backscatter LIDAR instruments from the 2014 DISCOVER-AQ field campaign in Colorado (low aerosol load) and coincident sonde launches from the Chemistry and Physics of the Atmospheric Boundary Layer Experiment (CAPABLE) site at NASA's Langley Research Center (LaRC; moderate aerosol load) in Hampton, Virginia.





## 2  Instrumentation

### 2.1  CL51

The Vaisala (Vantaa, Finland) CL51 ceilometer is a single-wavelength (eye safe Class 1M InGaAs diode laser emitting at 910 ± 10 nm, pulsed at 6.5 kHz with a 110 ns pulse width with average pulse power of 19.5 mW, and an avalanche photodiode detector centered at 915 nm), single-lens, LIDAR system originally designed to report cloud-base heights and visibility. More recently, ceilometers have been used to estimate MLH (Emeis and Schäfer, 2006; Emeis et al., 2008a, b; Haeffelin et al., 2012; Morille et al., 2007; Schäfer et al., 2012, 2013; Schween et al., 2014; Sokol et al., 2014; Wiegner et al., 2014). These ceilometers have 10 m vertical resolution (with 10 m overlap) to a maximum altitude of 15.4 km (± greater of 1% or 5 m precision, all altitudes are with respect to ground level) and up to 2 s temporal resolution (depending on the control software), though profiles are generally averaged over 16–36 s to improve the signal-to-noise ratio (see section 3.1 for more details). An example backscatter plot that includes increased signal at 3 km due to transport of smoke from a Canadian forest fire is presented in Fig. 1.

The CL51 was designed to operate continuously, regardless of meteorological conditions, in an autonomous manner with minimal user support. Due to the emission wavelength's proximity to the near-infrared water vapor bands these ceilometers experience water vapor interference, thereby mitigating their utility in retrieval of aerosol optical properties. However, the interference on aerosol profile and MLH estimation is negligible (Wiegner et al., 2014).

Two CL51s were deployed as part of the 2014 DISCOVER-AQ mission in Colorado (Golden, and Erie, Colorado). Before and after deployment, these ceilometers were set up to continually collect data at the CAPABLE site and the EPA Ambient Air Innovative Research Site (AIRS) in Durham, North Carolina. The ceilometers were collocated with meteorological sonde (met-sonde) launch sites during the DISCOVER-AQ campaign and at the CAPABLE site, allowing a direct intercomparison of the sonde and LIDAR ABL/MLH methodologies. Furthermore, during the DISCOVER-AQ campaign the ceilometers were collocated with other LIDAR instruments. Intercomparisons are presented in section 5.

### 2.1.1  Full-profile Collection

The Vaisala standard MLH retrieval is based on a proprietary wavelet/gradient technique built into the logging/analysis software BLView. The BLView software provides not only logging and data analysis (e.g. MLH and cloud-height estimates) but also archiving capability. While the CL51 reports backscatter up to 15.4 km, BLView truncates the data-collection at 4.5 km, precluding ability to monitor upper-troposphere/lower-stratosphere transport of aerosol, smoke, or ash from major events. Therefore, a full-profile collection method that can run side-by-side with the standard data-collection software was developed and implemented.

Data transmission from the ceilometer to the logging computer was achieved by splitting an RS-232 connection into two ports on the logging computer: one port logging to BLView and the other logging to a custom script (e.g. as written in Python). The primary drawback of using a secondary script to log the full profile (as opposed to logging in BLView) is the inability





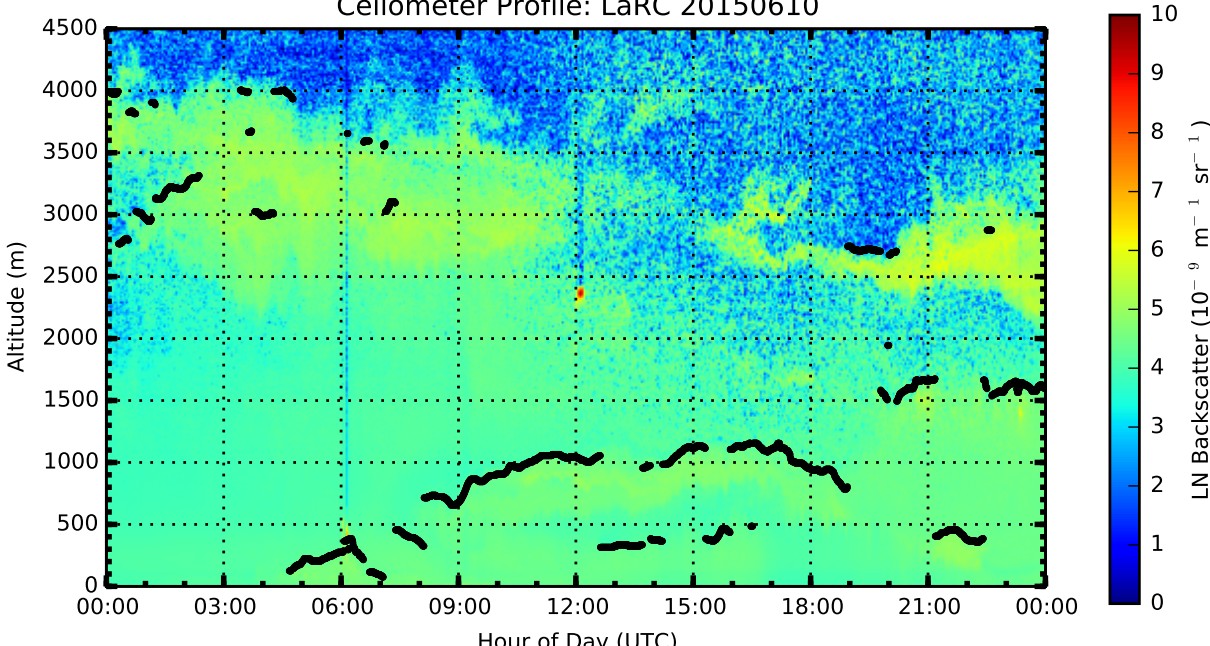

**Figure 1.** Backscatter curtain plot collected on 10-June 2015 when smoke from a Canadian forest fire was transported over the CAPABLE site. The smoke is observed by increased backscatter in the 2500 – 4000 m range.

to apply proprietary calibration coefficients that are built into the BLView software to the logged data. However, as shown in subsequent sections, this impacts neither the MLH estimates nor the general profile shape substantially.

## 2.2   Micropulse LIDAR

Elastic LIDAR observations were performed using a Sigma Space (Lanham, Maryland) Micropulse LIDAR (MPL), previously
5   described by Spinhirne (1993) and Welton et al. (2000). Briefly, the MPL transmitter consists of an eye-safe Nd:YLF laser emitting at 527 nm and pulsed at 2.5 kHz with a pulse power of $6 - 10$ $\mu J$. It has a software programmable vertical resolution, with possible values of 15, 30, and 75 m (up to 25 km), and temporal resolutions ranging from 1 s to 15 min. The receiver consists of a 178 mm telescope that collects the backscattered light, which is then focused onto a photon counting silicon avalanche photo-diode (APD). The APD output is recorded by a field programmable gate array data system that enables display
10   and storage of range dependent average count rates on a laptop computer. The raw data are converted to aerosol attenuated backscatter, correcting for instrumental factors such as detector dead time, geometrical overlap, background subtraction, and range-squared normalization. Recorded LIDAR profiles have temporal and vertical resolution of 1 min and 30 m, respectively. MPL is used for continuous recording of aerosol profiles and optical properties, and calculating MLH values.





## 2.3 Meteorological/Ozone Sondes

The meteorological sonde (herein referred to as sonde/radiosonde) is the conventional method of characterizing the ABL. Radiosondes were used to identify steep gradients within the potential temperature (theta) profile (Fig. 2 A) as identified by the Heffter criteria shown in Eqs. (1) and (2) where $\Theta$ is potential temperature in Kelvin, Z is altitude in meters, and $\Theta_{top}$ and $\Theta_{base}$ refer to the potential temperature at the top and bottom of the proposed inversion layer as described in (Heffter, 1980; Marsik et al., 1995). This thermodynamic ABL is a product of atmospheric turbulent kinetic energy and lapse rate. Similar gradients can be seen in chemical and aerosol profiles as well (Fig. 2 B-C). For the current study, radiosondes from International Met Systems (iMet; Grand Rapids, Michigan) and ozone sondes from Droplet Measurement Technologies (DMT, now En-Sci; Boulder, Colorado) were used. iMet sondes require no preparation and were used as received from the manufacturer, while ozone sondes were conditioned according to the procedure defined by the World Meteorological Organization recommendations (Smit, 2013).

$$\frac{\Delta\theta}{\Delta Z} \geq 0.005 K * m^{-1} \tag{1}$$

$$\Theta_{top} - \Theta_{base} \geq 2K \tag{2}$$

Results of numerous analyses have been published to illustrate differences between the various chemical and meteorological sensors, and how differing meteorological sensors influence secondary chemical measurements such as ozone (Deshler et al., 2008; Dirksen et al., 2014; Johnson et al., 2002; Miloshevich et al., 2004; Nash et al., 2006, 2011; Smit, 2013; Stauffer et al., 2014). While these influences can impact the derived CBL, the ABL and MLH remain unperturbed. Therefore, the remainder of the current work focuses on the MLH and ABL, with CBL variability regarded as outside the current scope.

## 3 Algorithms

### 3.1 BLView

BLView makes use of variable time and altitude averaging when calculating the MLH. Typical averaging time ranges from 14 min at night to 52 min during clear-sky, daytime conditions, and is automatically adjusted within the software according to signal-to-noise ratio. Altitude averaging varies with altitude and ranges from 80 m near the surface to 360 m above 1.5 km. Further, BLView selectively removes false-positive MLH identifications by requiring a minimum number of similar MLH values ($\pm$140 m) to be within the last several minutes, and has the ability to discriminate between MLH inversions and changes in backscatter intensity induced by cloud, precipitation, and fog.

Advantages of the BLView software are the standardization of retrieval parameters and a user interface that provides flexibility in setting user-specified sensitivities. These come at the cost of a database system that makes access to raw data difficult and the inability to batch process archived data, posing a severe limitation on reprocessing datasets with a long record history.





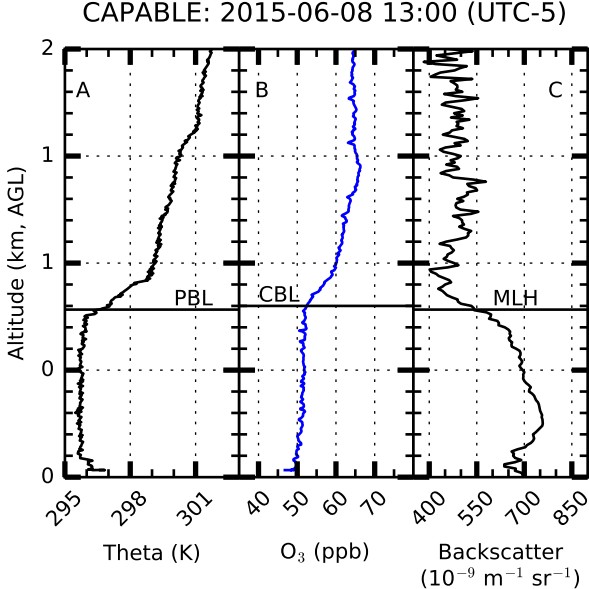

**Figure 2.** Potential temperature, ozone, and backscatter profiles recorded on 8-June 2015. The horizontal lines indicate the ABL, CBL, and MLH.

## 3.2 STRAT

The STRucture of the ATmosphere (STRAT v1.04) algorithm was developed under a GNU General Public License to analyze aerosol vertical profiles as measured by LIDAR and estimate cloud heights and aerosol MLH from a variety of LIDAR instruments. It is currently in use by the European Aerosol Research Lidar NETwork (EARLINET) (Haeffelin et al., 2012; Hirsikko et al., 2014; Morille et al., 2007; Pappalardo et al., 2014). STRAT uses a covariance wavelet technique (CWT), of which the full details can be found in Morille et al. (2007) and Haeffelin et al. (2012). STRAT can be run exclusively in MATLAB, or a combination of MATLAB and Python. Due to its wide use throughout the European network it is considered here as a viable open-source alternative to BLView.

While BLView provides limited user control of the retrieval process, which is beneficial with regard to standardizing the retrieval process across a network, STRAT provides a significantly greater amount of user control. Such control is desirable since retrieval parameters in a heavily polluted region will likely be different from those in a clean environment. Further, STRAT is provided as raw scripts as opposed to BLView's compiled executable, making the STRAT platform independent and highly user-configurable. STRAT also can run batch jobs, which is useful when reprocessing data from instruments that have a long record history.





The STRAT algorithm implements a user-defined normally-distributed weighting function in both the temporal and vertical domains to smooth the data, similar to BLView. In the current study, the STRAT parameters were set to match the BLView settings as much as possible for intercomparison. An analysis of how well the two MLH algorithms agree is presented below.

### 3.3 UMBC Algorithm

5 The University of Maryland Baltimore County (UMBC) algorithm was developed independently for estimating MLH from LIDAR backscatter profiles using a CWT similar to STRAT. The STRAT software was designed specifically for single-channel LIDARs (primarily ceilometers) and is not readily customizable to other LIDAR systems, such as the MPL. The UMBC algorithm was designed to be more flexible than STRAT in that regard and uses a CWT to identify the sharp gradient changes indicative of the MLH (Davis et al. 2000; Brooks 2003).

10 Detailed description of the UMBC algorithm has been published elsewhere Compton et al. (2013). The first step in the UMBC algorithm defines the dilation and center of the Haar function values considered in the CWT. The second step consists of applying the CWT to the LIDAR profile for the appropriate dilation and center of the Haar function values. The sharp gradients in the profile that are of interest are identified by local minima in the resulting wavelet covariance profile. The local minimum is selected as the MLH, and the process is repeated for each profile in the data set.

### 4 Locations

#### 4.1 CAPABLE Site

The CAPABLE site was established at LaRC, in the greater Hampton Roads region (a group of cities in coastal Virginia, also known as Tidewater Virginia: Virginia Beach, Norfolk, Chesapeake, Newport News, Hampton, Portsmouth, Suffolk, Poquoson, Williamsburg), for continuous monitoring of air-quality and meteorological parameters to bridge the gap between satellite 20 observations and ground conditions (i.e., where pollutants directly impact living organisms), improve applicability of satellite data to the air-quality user community, and act as a long-term satellite validation site. CAPABLE has a suite of in situ and remote-sensing instruments, including a CL51 ceilometer and sounding station. These instruments allow thorough sampling of the atmosphere to provide valuable in situ and profile information within the lower troposphere in a highly complex (due to bay-breeze events; see Martins et al. (2012)) and moderately polluted ($NO_x$, $SO_2$, aerosols) environment yielding valuable 25 satellite ground-truthing and model a priori estimates.

CAPABLE (37.103º N, 76.387º W, 5 m ASL) is located on a peninsula between the James River to the southwest, the Chesapeake Bay to the north, and the Atlantic Ocean to the east. Immediate emission sources and their locations relative to CAPABLE are as follows: commuter traffic (Wythe Creek Rd to the west at ≈15,000 vehicles per day and Commander Shepard Blvd to the south at ≈20,000 vehicles per day; Commander Shepard and Wythe Creek share much of the same traffic, so it is not reasonable to estimate a total traffic flow of 35,000 vehicles per day); Yorktown Power Station (approximately 350





| Site | $\lambda$ (nm)/PM Size | Mean | $Q_1$ | $Q_2$ | $Q_3$ |
|---|---|---|---|---|---|
| BAO-Tower | 380 | 0.23 | 0.13 | 0.19 | 0.32 |
| BAO-Tower | 500 | 0.15 | 0.09 | 0.13 | 0.22 |
| BAO-Tower | 675 | 0.09 | 0.05 | 0.08 | 0.13 |
| BAO-Tower | 870 | 0.06 | 0.04 | 0.06 | 0.08 |
| BAO-Tower | 1020 | 0.05 | 0.03 | 0.04 | 0.06 |
| Golden, CO | 380 | 0.20 | 0.10 | 0.16 | 0.27 |
| Golden, CO | 500 | 0.13 | 0.06 | 0.10 | 0.18 |
| Golden, CO | 675 | 0.08 | 0.04 | 0.06 | 0.11 |
| Golden, CO | 870 | 0.05 | 0.03 | 0.04 | 0.07 |
| Golden, CO | 1020 | 0.04 | 0.02 | 0.03 | 0.05 |
| CAPABLE | 380 | 0.34 | 0.21 | 0.33 | 0.45 |
| CAPABLE | 500 | 0.23 | 0.13 | 0.23 | 0.32 |
| CAPABLE | 675 | 0.14 | 0.08 | 0.14 | 0.20 |
| CAPABLE | 870 | 0.09 | 0.05 | 0.09 | 0.13 |
| CAPABLE | 1020 | 0.07 | 0.03 | 0.06 | 0.10 |
| CAPABLE | $PM_{2.5}$ | 5.80 | 2.59 | 5.00 | 8.44 |

**Table 1.** Aerosol optical thickness statistics at the three sites under study. Here, $Q_1$, $Q_2$, and $Q_3$ represent the $25^{th}$, $50^{th}$, and $75^{th}$ percentiles, respectively. Data have been filtered to show only data collected during the DISCOVER-AQ 2014 field campaign period (July – August 2014).

MW, 1150 MW peak) and Yorktown oil refinery to the north-northwest; Langley Air Force Base to the southeast; Richmond, Virginia to the west; and Baltimore and Washington DC, further to the north.

The Hampton Roads region can be described as moderately polluted. Aerosol statistics ($PM_{2.5}$ and aerosol optical thickness (AOT) as recorded by a sun photometer within the AERosol Robotic NETwork (AERONET) as described by Holben et al.
(1998)) are presented in Table 1. The data show AOT loads at CAPABLE are significantly higher than at the corresponding Colorado sites, particularly in the lower size distributions (i.e., lower wavelengths in Table 1).

## 4.2 DISCOVER-AQ Sites

From 2011 through 2014 the National Aeronautics and Space Administration (NASA) conducted the Deriving Information on Surface Conditions from Column and Vertically Resolved Observations Relevant to Air Quality (DISCOVER-AQ) Earth Venture Suborbital Mission with four field deployments: Baltimore/Washington region of Maryland during 2011; the San Joaquin Valley (SJV) of California during January and February 2013; Houston, Texas during September 2013; and the Front Range region of Colorado in July and August 2014. A primary objective of DISCOVER-AQ was to investigate the ability of satellite remote sensing to inform air quality at the surface. Since the ABL limits vertical exchange of primary pollutants and directly influences near-surface pollutant concentrations, the ABL height can directly influence air quality and chemistry.



Therefore, measurements during these missions focused on the vertical distribution of trace gases and aerosols within the ABL and lower troposphere, as well as the diurnal variability of these distributions in conjunction with the ABL.

### 4.2.1 Erie, Colorado/BAO-Tower

Data were collected at the Erie, Colorado site (40.045º N, 105.005º W, 1500 m ASL), which is considered to be a clean
environment as compared to CAPABLE (see Table 1) from 14-July to 12-August 2014 as part of the DISCOVER-AQ field mission. The Erie site was located at NOAA's Earth System Research Laboratory's (ESRL) Boulder Atmospheric Observatory (BAO) in Erie, Colorado, a rural community surrounded by agricultural activity. The site is often called the BAO-Tower because of the site's primary feature: a 300 m tower. BAO-Tower provided a unique profiling ability for in situ samplers by mounting them on the tower for static sampling, or on the carriage to collect "active" profiles.

During DISCOVER-AQ 2014, the University of Wisconsin's (UW) Space Science and Engineering Center trailer, which housed a high spectral resolution lidar and from which regular sonde launches were performed, was stationed at the site. The UW trailer temporarily housed a CL51 during the mission. Due to the proximity of the UW trailer, both ceilometers experienced the same chemical, aerosol, and meteorological conditions.

### 4.2.2 Golden, Colorado

Data were collected at the Golden, Colorado site (39.750º N, 105.183º W, 1850 m ASL) (considered to be a clean environment as compared to CAPABLE, see Table 1) from 14-July to12-August 2014 as part of the DISCOVER-AQ field mission. The Golden site was located next to the National Renewable Energy Laboratory (NREL) on Table Mountain mesa. Due to the site's elevation on the mesa, and its limited emissions sources, conditions at the Golden site were generally clean from an aerosol perspective and did not typically experience a well-developed ABL/ML.

The Golden site housed the U.S. EPA trailer, the LaRC ozone LIDAR, MPL and LEOSPHERE ALS-450 LIDAR operated by UMBC, a SOnic Detection and Ranging (SODAR) instrument operated by Millersville University (MU), and regular met-sonde launches from the MU group.

## 5   Analysis

LIDAR data collected during the DISCOVER-AQ campaign had sampling times that ranged from 36 to 60 s, while sonde-
profile data had average measurement times of 1 s. To harmonize all datasets to a common time frame the data were averaged to 5 min resolution unless otherwise specified. Further, it is well known that the atmosphere changes throughout the day due to surface heating, etc. (hence, driving ABL variability). Therefore, some of the analyses were broken into 4-hour segments to remove biases caused by time-of-day influences. Since the primary objective of this assessment was to understand how the CL51 MLH compared with other instruments/methods, all analytical results are presented in relation to the CL51.
The analysis was performed using several ceilometer MLH products to do a thorough comparison of instruments (CL51, MPL, and met-sondes), collection method (allowing BLView to collect profile data with application of calibration factors vs.




logging raw data with a custom Python script), and data processing algorithm (BLView vs. STRAT and custom MLH scripts from UMBC). Assessment of data-acquisition methodology is presented first, followed by a comparison of MLH retrieval algorithms applied to data collected by a single instrument, and then a comparison of the various instrumentation.

## 5.1 Data Acquisition

Data-acquisition methods were analyzed to determine whether the CL51 data-logging methodology influenced the MLH estimate. As described above, CL51 profile data were logged using two methodologies: BLView and a custom Python routine. The BLView software has the advantage of applying the ceilometer's calibration factors and preconditioning the profiles (here referred to as BLView; note, however, that this refers to the backscatter-profile that is logged by BLView and not the BLView-calculated MLH), while the Python script logged the raw incoming data stream up to the full profile (FP) height (i.e., 15.4 km).

The question was, does application of the LIDAR calibration factor influence the MLH estimate? This question is addressed in section 5.1.2, but first, viable filtering criteria to remove spurious MLH fluctuations from the data set were developed prior to analysis, as discussed in section 5.1.1.

### 5.1.1 Filtering Criteria

Regardless of the data-acquisition method (i.e., BLView or Python), pragmatic data-selection criteria were needed for quality

control. Since ABL and MLH variations occur in a generally smooth manner, it is expected that the variance within a short time interval will be minimal, and that any larger variance is indicative of other events (e.g. precipitation, window contamination). Therefore, cutoff criteria for implementing data filtering were identified. This portion of the analysis was conducted first because application of these cutoff criteria will influence the data acquisition comparison (i.e., BLView-corrected data vs. raw data collected via the Python script).

Despite the atmosphere's smooth variation in ABL and MLH, these parameters do change substantially over long periods of time (e.g. an hour or day), with standard deviation significantly increased over the longer time periods. Therefore, the current analysis was performed on short-time-series data (i.e., 5 min) to eliminate bias caused by natural low-frequency changes. Figure 3 shows a series of percentile plots for data collected at LaRC (N > 30E5), where the standard deviation was calculated over 5-min intervals. This figure elucidates the variability of the MLH standard deviation for both collection methods and algorithms.

Except for the afternoon period (12:00 – 19:00) local time when the variability is slightly increased, 85% of the data fall within one standard deviation ($\approx 0.20$ km) regardless of time of day. Therefore, data with a 5-min standard deviation greater than 0.20 km were removed from subsequent analysis (labeled "filtered"). Data with a relative standard deviation greater than or equal to 20% were also removed. Implementation of these filter criteria remove up to 10% of the data.

This filtering method is further supported by observing the variability in the BLView and Python-collected datasets (both

processed in STRAT) in relation to backscatter curtains (Fig. 4) where it is observed that much of the difference between the BLView and Python-collected data occurs during times of high variability or precipitation (e.g. 19:00 – 24:00 in Fig. 4). During such events, neither collection method is expected to provide valid MLH estimates; rather, to overcome such discrepancies, if possible, the MLH algorithms must be adjusted accordingly.





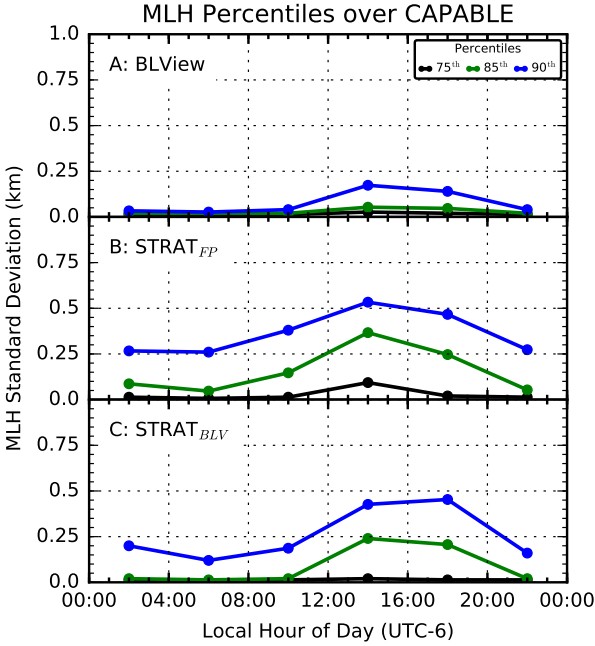

**Figure 3.** Percentiles for MLH standard deviation throughout the day from the CAPABLE site. Data in panel A were collected and processed in BLView, data in panel B collected with the Python script and processed in STRAT, data in panel C were collected in BLView and processed in STRAT. It is observed that variability was maximum during the afternoon regardless of collection method or processing algorithm.

### 5.1.2 Collection Method Dependence

To determine whether the data-collection method influenced MLH estimates, both BLView and Python-collected backscatter profiles were processed on a common algorithm (STRAT) using identical input configuration files. Both the BLView and FP profiles were processed using the STRAT algorithm as described in section 3.2, followed by a 5-min block average. Figure 5 presents relative standard deviation as a function of distance (binned in 50 m increments) from the 1:1 line (i.e. absolute difference between the two methods) within the 5-min interval. It is observed that the relative standard deviation increased rapidly up to ≈300 m and slowly decreased with increasing distance. This is further support for the selection of filtering criteria.

The data were replotted as correlation plots with the $z$-axis being representative of the immediate data density (a dimensionless value that has been scaled to 1). The data density was calculated by implementing a Gaussian-based kernel-density estimation (Scott, 1992; Silverman, 1986) as supplied in Python's scipy.stats.kde module, represented mathematically in Eqs. 3–5 where $\mathbf{X}$ is the 2 x $n$ vector of the $\mathbf{x}$ and $\mathbf{y}$ vectors (i.e., flattened and stacked atop one another), $n$ represents the number of points within each dataset (assuming datasets are of equal length), $f$ is the Scott's factor ($n^{\frac{-1}{d+4}}$), $d$ is the number of independent datasets analyzed, and Eq. 5 is evaluated over the range 1 to n. As these density values are used as weights in subsequent



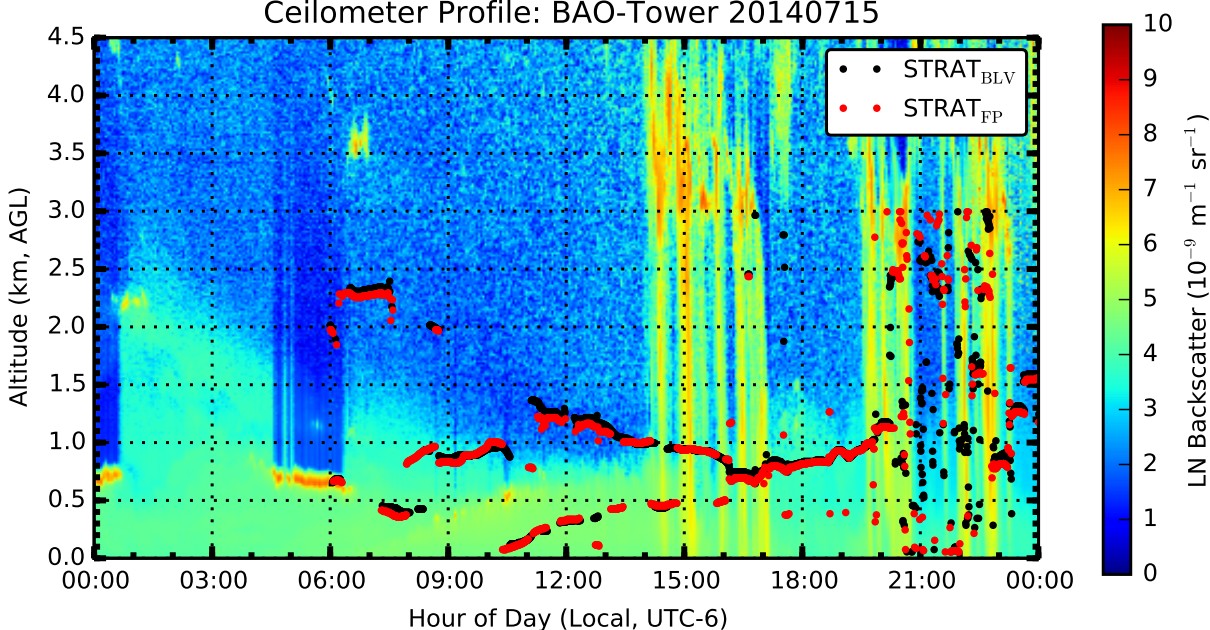

**Figure 4.** Backscatter curtain plot with STRAT-derived MLH values from the BLView (BLV) and Python (FP) collection methods.

calculations, the output vector is labeled $\mathbf{w}$ here. It is observed that the majority of MLH estimates fall along the 1:1 line (center column in Fig. 6), though there is significant scatter along both axes.

$$\Delta \mathbf{X} = \mathbf{X} - \mathbf{X}[:,i] \tag{3}$$

$$\mathbf{E} = \sum_{j=1} \Delta \mathbf{X}_j \cdot \frac{cov(\mathbf{X})^{-1}}{f^{-2}} \bullet \Delta \mathbf{X}_j \tag{4}$$

5 $$\mathbf{w}[i] = \left. \frac{\sum\limits_{k=1} e^{-\mathbf{E}_k}}{\sqrt{det\left[2\pi \cdot cov(\mathbf{X}) \cdot f^2\right]}} \right|_i^n \qquad \{i \in \mathbb{N}: i \leq n\} \tag{5}$$

Figure 6 was divided into 4-hour blocks to identify any time-of-day dependence. The figure shows that most of the data continued to fall along the 1:1 line regardless of time of day, as indicated in the CAPABLE and BAO-Tower density plots. The Golden site displays some disruption in the $16:00 - 19:59$ panel, but the source of this discrepancy is currently unknown. It has become clear, however, that the meteorology at the Golden site is different from that observed at CAPABLE and BAO-Tower.

10 It is suggested that this difference is primarily driven by orographic perturbations as well as the Golden site's location atop a





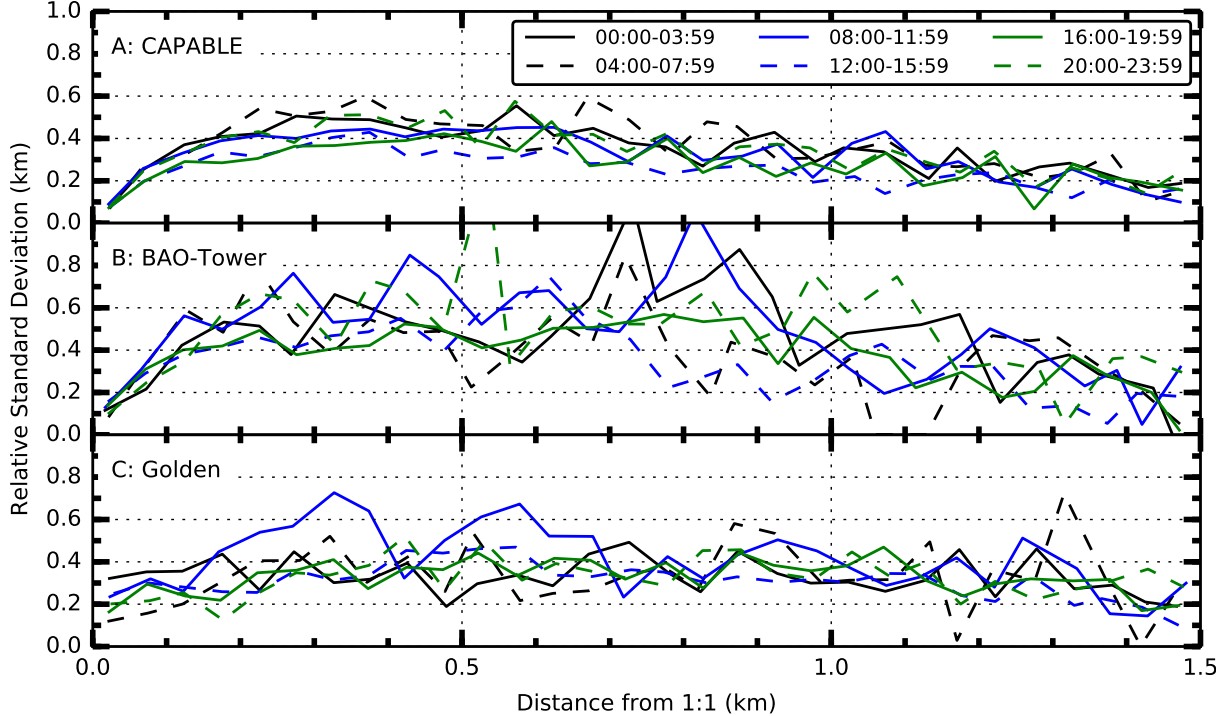

**Figure 5.** Relative standard deviation ($\sigma/\bar{x}$) as a function of distance from the 1:1 line. Data were binned in 50 m bins (distance from 1:1) and subsequently averaged.

mesa, both of which can inhibit formation of stable ABL and ML (Bossert et al., 1989; Bossert and Cotton, 1994; Tripoli and Cotton, 1989).

For regulatory and modeling applications, 1-hour averages are standard, requiring the data be averaged down to 1-hour resolution. The impact of the filtering criteria and re-sampling to 1-hour resolution throughout the day can be seen in Fig. 7.

5  Table 2 presents statistics on the aggregate analysis. While the aggregate coefficients of correlation and line-of-best-fit (LOBF) equations do not change substantially after re-sampling to 1-hour blocks, the scatter is dramatically reduced. This is likely due to the scatter being evenly distributed around the 1:1 line and the majority of data points falling along the 1:1 line, as observed in the data-density panels of Fig. 7.

It can be concluded from the current analysis that the majority of variability was driven by local atmospheric fluctuations and

10  events that cannot be readily accounted for within the algorithms. In addition, no significant difference is observed between the BLView- and Python-collected data sets on the timescales relevant to model inputs and atmospheric variations, when processed on a common algorithm. Findings presented in section 5.1.3 further support this conclusion.





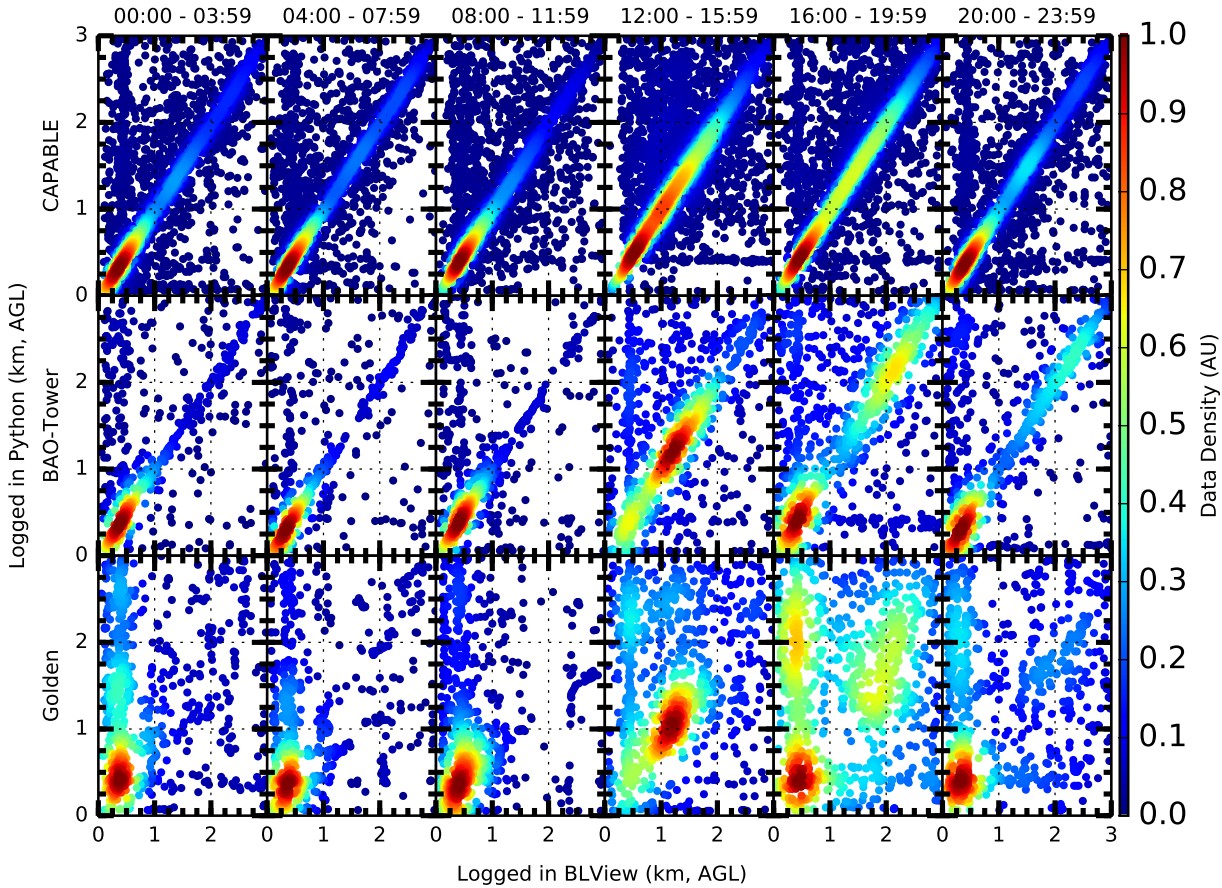

**Figure 6.** Correlation plots for data collected at the three sites under study. Data density is presented to better understand the distribution within the scatter plots. Data were averaged to 5-min resolution, without application of filtering criteria.

|  | R | LOBF | $\langle FP - BL \rangle$ (km) |
|---|---|---|---|
| CAPABLE 5-min | 0.87 | $y = 0.913 \cdot x + 0.11$ | $-0.02$ (1.4) |
| CAPABLE 1-h | 0.87 | $y = 0.925 \cdot x + 0.11$ | $-0.03$ (2.7) |
| BAO 5-min | 0.76 | $y = 0.817 \cdot x + 0.25$ | $-0.08$ (9.1) |
| BAO 1-h | 0.77 | $y = 0.814 \cdot x + 0.32$ | $-0.14$ (15.1) |
| Golden 5-min | 0.72 | $y = 0.777 \cdot x + 0.30$ | $-0.08$ (8.1) |
| Golden 1-h | 0.77 | $y = 0.792 \cdot x + 0.35$ | $-0.14$ (13.0) |

**Table 2.** Summary of aggregate statistics for the Python-collected (FP)/STRAT-processed and the BLView-collected (BLV)/STRAT-processed MLH estimates ($y$ and $x$, respectively). Filtering criteria were applied to both datasets. Values in parentheses indicate percent of the difference value with respect to the BLView-derived MLH.





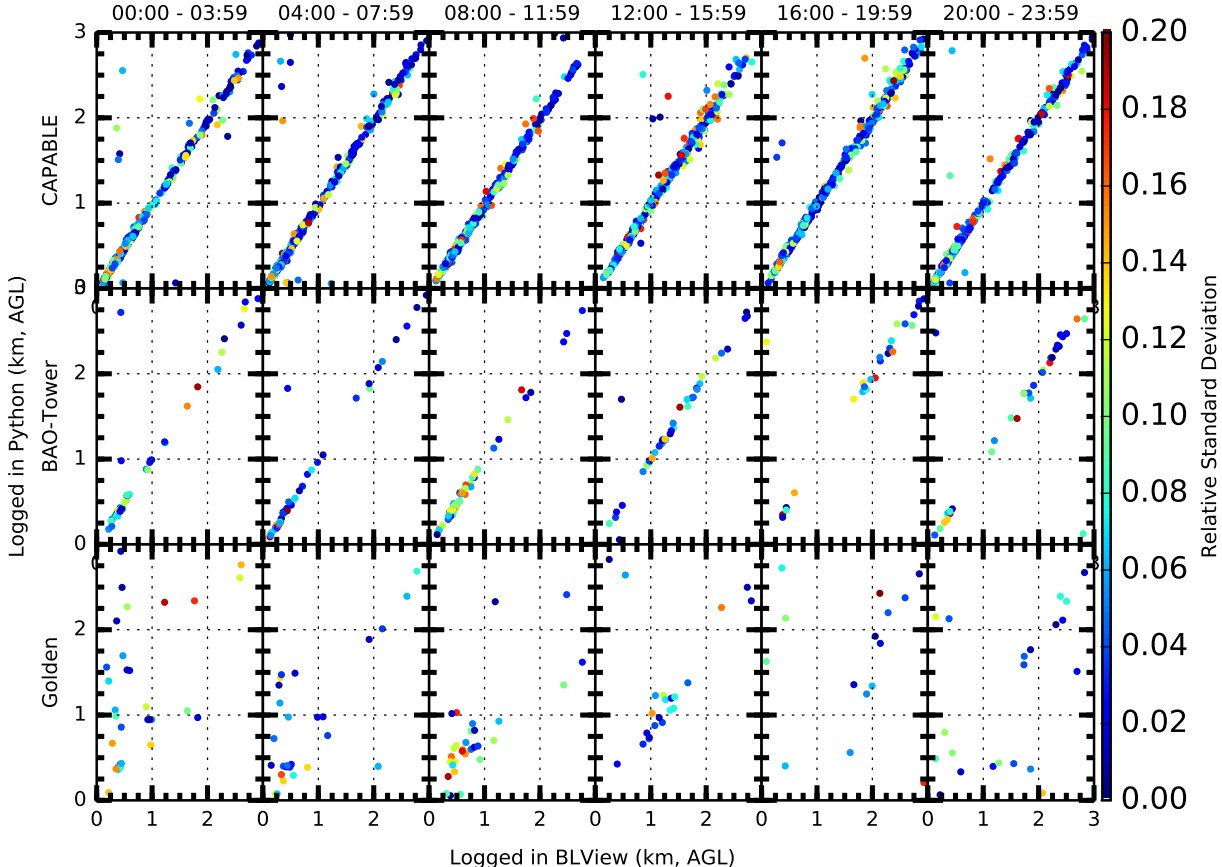

**Figure 7.** Same as in Figure 6, but with the data resampled to 1 h means and application of filtering criteria.

### 5.1.3 MLH Algorithm Dependence

In the previous section, the data collection method (i.e., Python vs. BLView) was shown to have little impact on the derived MLH values when the two datasets were processed using a common algorithm (STRAT). The question remains of how the two data sets compare when processed in different algorithms. To answer this question, data collected with the Python script were processed using the STRAT algorithm and were compared with data collected and processed with BLView.

Figure 8 presents scatter plots similar to those in Fig. 6, but with data collected and processed using the two different methods. Most data continued to fall along the 1:1 line, as shown in the density plots, and much of the scatter is caused by short-term variability. However, in contrast to Fig. 6, the scatter is neither as evenly distributed nor as tightly grouped around the 1:1 line. The STRAT-derived MLHs were generally lower than those calculated in BLView (given by the slopes) at all sites, while the aggregate mean difference shows the opposite for the Colorado sites (Table 3), which is likely driven by outliers.



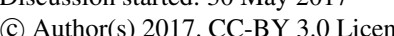


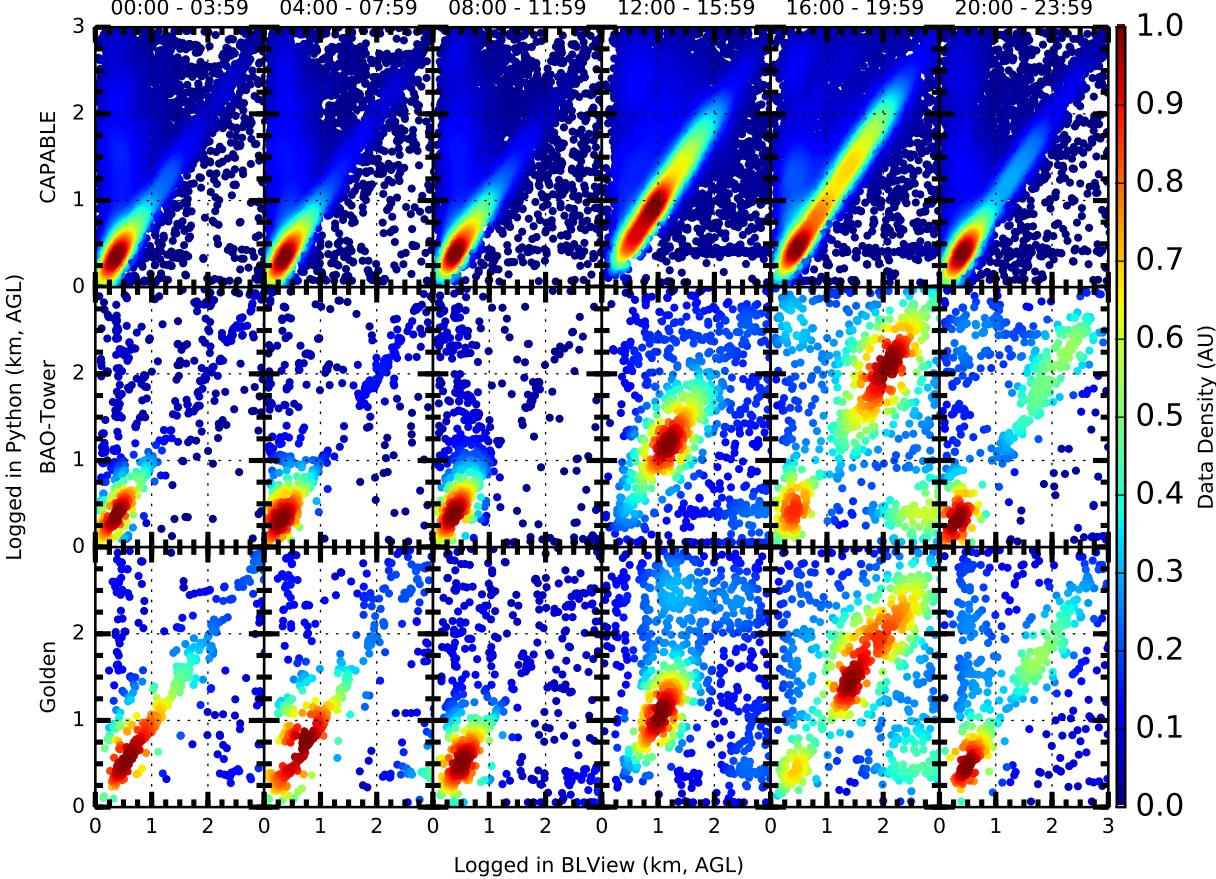

**Figure 8.** Correlation plots for data collected at the three sites under study. At all sites the data were collected by, and processed in, Python/STRAT and BLView/BLView. Plots show the data density to better understand the distribution within the scatter plots. Data were averaged to five-minute resolution, without application of filtering criteria.

The agreement between the two datasets is less than when a common algorithm was employed (Table 3). Despite the increased scatter, a significant subset of data remains along the 1:1 line. As a test for how well the data fit the 1:1 line, the R and LOBF values were re-calculated using Eq. 5 with weights applied according to data density. Therefore, points that had a greater number of surrounding data points received more weight, while more isolated points received less weight. Weighted

5    coefficients of correlation were calculated using Eq. (6), where variables with a $w$ subscript indicate weighted means. Weighted




|  | R | Line of best fit | $\langle FP - BLV \rangle$ (km) | $R_w$ | $LOBF_w$ | $\langle FP - BLV \rangle_w$ (km) |
|---|---|---|---|---|---|---|
| CAPABLE 5-min | 0.47 | $y = 0.499 \cdot x + 0.70$ | $-0.24\ (26.8)$ | 0.836 | $y = 0.986 \cdot x + 0.70$ | $-0.02\ (3.8)$ |
| CAPABLE 1-h | 0.48 | $y = 0.467 \cdot x + 0.74$ | $-0.23\ (24.4)$ | 0.799 | $y = 0.997 \cdot x + 0.74$ | $-0.08\ (12.5)$ |
| BAO 5-min | 0.43 | $y = 0.374 \cdot x + 0.65$ | $0.04\ (3.4)$ | 0.789 | $y = 0.905 \cdot x + 0.65$ | $-0.01\ (0.9)$ |
| BAO 1-h | 0.39 | $y = 0.305 \cdot x + 0.72$ | $0.17\ (13.1)$ | 0.740 | $y = 0.881 \cdot x + 0.72$ | $-0.01\ (1.2)$ |
| Golden 5-min | 0.24 | $y = 0.193 \cdot x + 0.90$ | $0.25\ (17.7)$ | 0.541 | $y = 0.629 \cdot x + 0.90$ | $0.09\ (10.3)$ |
| Golden 1-h | 0.12 | $y = 0.086 \cdot x + 1.12$ | $0.39\ (23.6)$ | 0.316 | $y = 0.361 \cdot x + 1.12$ | $0.20\ (16.1)$ |

**Table 3.** Summary of statistics for the Python-collected/STRAT-processed and the BLView-collected/BLView-processed MLH estimates. Values in parentheses indicate percent of the difference value with respect to the BLView-derived MLH, and the $w$ subscript indicates a weighting function was applied. Filtering criteria were applied to all datasets.

|  | R | Line of best fit | $\langle FP - BLV \rangle$ (km) | $R_w$ | $LOBF_w$ | $\langle FP - BLV \rangle_w$ (km) |
|---|---|---|---|---|---|---|
| CAPABLE 5-min | 0.54 | $y = 0.553 \cdot x + 0.61$ | $-0.19\ (21.2)$ | 0.91 | $y = 0.975 \cdot x + 0.61$ | $-0.03\ (4.2)$ |
| CAPABLE 1-h | 0.54 | $y = 0.519 \cdot x + 0.64$ | $-0.18\ (18.5)$ | 0.87 | $y = 0.973 \cdot x + 0.64$ | $-0.06\ (9.8)$ |
| BAO 5-min | 0.41 | $y = 0.326 \cdot x + 0.58$ | $0.14\ (13.1)$ | 0.78 | $y = 0.843 \cdot x + 0.58$ | $-0.01\ (1.4)$ |
| BAO 1-h | 0.31 | $y = 0.232 \cdot x + 0.65$ | $0.32\ (25.5)$ | 0.58 | $y = 0.573 \cdot x + 0.65$ | $-0.09\ (13.0)$ |
| Golden 5-min | 0.25 | $y = 0.184 \cdot x + 0.83$ | $0.36\ (24.5)$ | 0.49 | $y = 0.484 \cdot x + 0.83$ | $0.16\ (16.6)$ |
| Golden 1-h | 0.14 | $y = 0.101 \cdot x + 0.96$ | $0.55\ (32.8)$ | 0.27 | $y = 0.229 \cdot x + 0.96$ | $0.37\ (29.1)$ |

**Table 4.** Summary of statistics for the BLView-collected/STRAT-processed and the BLView-collected/BLView-processed MLH estimates. Values in parentheses indicate percent of the difference value with respect to the BLView-derived MLH, and the $w$ subscript indicates a weighting function was applied. Filtering criteria were applied to all datasets. Herein, the comparison is limited strictly to the MLH algorithms.

regressions were performed by simultaneously solving the modified normal equations of regression shown in Eqs. (7) and (8) with weighting factors applied.

$$R = \frac{\sum_{i=1}^{N} w_i \cdot (x_i - \bar{x}_w) \cdot (y_i - \bar{y}_w)}{\sqrt{\sum_{i=1}^{N} w_i \cdot (x_i - \bar{x}_w)^2 \cdot \sum_{i=1}^{N} w_i \cdot (y_i - \bar{y}_w)^2}} \tag{6}$$

$$m = \frac{N \sum_{i=1}^{N} w_i x_i y_i - \left( \sum_{i=1}^{N} w_i x_i \right) \left( \sum_{i=1}^{N} w_i y_i \right)}{N \sum_{i=1}^{N} w_i x_i^2 - \left( \sum_{i=1}^{N} w_i x_i \right)^2} \tag{7}$$





$$b = \frac{\sum_{i=1}^{N} w_i y_i - m \sum_{i=1}^{N} w_i x_i}{N} \qquad (8)$$

These weighted statistics are not included to suggest that the agreement has actually improved (R), nor do they suggest improved predictability (LOBF). Rather, the improved R values and slopes reflect the degree to which the data are predominantly distributed around the 1:1 line to the exclusion of other regions. As an example, the improvement in the Golden regressions,

despite weighting, is notably less than the other two sites. This is likely due to more spread in the data, which mitigates the influence of the points along the 1:1 line in the regression analyses. Therefore, the preponderance of the data collected at the CAPABLE and BAO-Tower sites fall nearer the 1:1 line when processed using the different algorithms as compared to the data collected at the Golden site. Further, despite most data falling nearer the 1:1 line for these two sites, influences remain that neither the STRAT configuration nor the current filter methodology can account for, which is likely driving the poor correlation

as compared to Table 2. This is possibly a product of how the differing algorithms handle atmospheric interferential events (e.g. precipitation, fog). Application of a filtering methodology to account for and remove these events will be the subject of future study.

Finally, the analysis was repeated by using STRAT to process backscatter data collected by BLView for comparison with the BLView-collected/processed product. As concluded in section 5.1.2, the data collection method had little influence on the

MLH estimation when both datasets were processed using a common algorithm (STRAT). Based on that conclusion, it would be expected that the current comparison would be similar to the previous comparison as summarized in Table 3. This is, in fact, what was observed. The aggregate statistics for the BLView-collected, STRAT-processed vs. BLView-collected/processed intercomparison are presented in Table 4, wherein we see similarity with Table 3. These findings further support the conclusion that data collection methods (including application of calibration factors) play much less of a role in identifying a qualitative

gradient within the profile than the choice of MLH algorithm. Indeed, it can be concluded that choice and configuration of the algorithm is critical and that, for network intercomparisons, all networked LIDAR systems should have their data processed by a common algorithm.

## 5.2   Sonde Intercomparison

Meteorological soundings have been a staple for profiling the atmosphere and deriving ABL heights for decades. These ABL

heights are typically derived using potential temperature (e.g. using the Heffter criteria) or through analyzing skew-T, log-P plots that implement potential temperature, both of which are different from the gradient-based MLH algorithms implemented here. As ABL data are typically used in chemical transport models, it is necessary to determine how these MLH data compare to the sonde-derived ABL data collected at the three measurement locations.

Since the sondes capture an ephemeral snapshot of the atmosphere's current conditions and traverse several kilometers

in the horizontal direction due to winds, the ceilometer data were averaged over 30-min for comparison. Additionally, each measurement can be impacted by atmospheric phenomena that can affect the measurements in different ways and can in turn



affect the comparison of the measurements. Met-sondes can be impacted by local updrafts or downdrafts and result in ABL estimates that are higher or lower than the time- or space-averaged MLH. The CL51 MLH is calculated based on identification of a sufficiently steep, vertically-averaged, backscatter gradient, so if there are additional aerosol layers just above the MLH, the contrast between the aerosol layers might not be strong enough for the CL51 to identify each layer or the correct altitude

of the MLH.

Correlation plots for the CL51 MLH compared to sonde ABL are shown in Fig 9. For all coincidence times, the CAPABLE site showed the best correlations between the CL51 and sondes. The correlation for the CL51 versus all the sondes (N = 25) at the CAPABLE site was R=0.82, with a similar correlation R = 0.83 (N = 22) when the filtering criteria were implemented. For daytime data, the CAPABLE site contained two early morning sondes (before 10:00 local time), with all other sondes launched

between 10:00 and 16:00 local time. By late morning, ≈10:00 local time, the vertical dispersion of aerosols due to turbulent mixing has likely resulted in a well-mixed boundary layer, so the ABL and MLH coincide in elevation, which is evident in Fig. 9 A where many of the data points fall close to the 1:1 line.

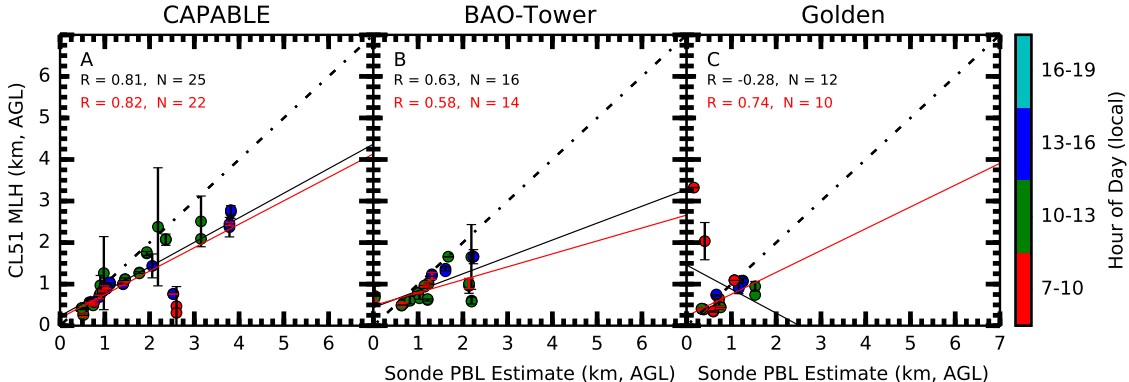

**Figure 9.** Correlation plots for CL51 MLH and sonde-derived ABL estimates. Statistics data in black text are for the entire data set, while the red text represents the filtered dataset. MLH values were calculated in BLView.

Met-sonde data collected at the BAO-Tower site showed lower correlations than the CAPABLE site (unfiltered R = 0.63, N = 16; filtered R = 0.58, N = 14), while the Golden site correlations (unfiltered R = -0.28, N=12) appear to be strongly impacted

by two morning sonde launches, which occurred during a transition period when the boundary layer was experiencing rapid growth. Upon applying the filtering criteria, the two early morning data points were removed, resulting in a much improved correlation (filtered R = 0.74, N = 10) for the Golden site. These results indicate the CL51 might have difficultly capturing an accurate MLH during rapidly changing conditions, such as during early morning and late evening transition periods in a clean atmosphere.

It is somewhat surprising that filtered correlation for the Golden site is better than the filtered result for the BAO-Tower site, given the BAO-Tower site is situated farther to the east of the Rocky Mountains, at the start of the High Plains, which are less influenced by very local geographic perturbations, and that a similar relationship is not observed in the CL51 intercomparisons





(Tables 2, 3, and 4). As a check of the met-sonde potential temperate profiles, the potential temperature data from the NASA P-3B aircraft spirals conducted over the Golden and Erie sites is shown in Figs. 10 and 11. These spirals are coincident with the launch of the met-sondes from the sites. The coincident CL51 backscatter profiles are also plotted in Figs. 10 and 11. The agreement between the radiosonde and P-3B aircraft profiles is good, indicating that the potential temperature within the

aircraft spiral radius is consistent with that of the radiosonde. These figures show agreement between the potential temperature ABL and CL51 MLH by identifying the same first major gradient in the MLH data on certain days.

Overall, all three sites show good correlation between the CL51 and met-sonde data, with MLH and ABL estimates from the sondes being, on average, higher than the CL51 MLH (200 m (13%), 390 m (15%), -240 m (9%) for CAPABLE, BAO-Tower, and Golden respectively) as indicated in the linear regression lines plotted in Fig. 9, with the exception being the unfiltered

results for Golden.

### 5.3 MPL Intercomparison

The MPL instrument was collocated with the CL51 stationed at the NREL site in Golden, Colorado. Being a LIDAR instrument, it profiles the atmosphere similarly to the CL51 with the major difference being their hardware. The two instruments emit different wavelengths (CL51:910 nm, MPL:532 nm), causing the instruments to differ in sensitivity with respect to particle

size and geometry. Therefore, it is feasible that the two instruments observed "different" atmospheres in a quantitative manner (e.g. AOT). However, if the ML is well mixed, then the general particle distribution and gradient will be the same, making the two inter-comparable.

Figure 12 shows that the agreement between the two instruments and algorithms (BLView processing CL51, UMBC algorithm processing MPL profiles) is poor, even though a significant subset of data fall along the 1:1 line, as indicated by data

density ($z$-axis). The low correlation is partly driven by the invariability in one instrument as compared to the other at lower MLH values ($\leq 500$ m). Removal of data below 500 m improved the coefficients of correlation for the 5-min averaged data to 0.368, 0.512, and 0.390 respectively. Similar to the algorithm comparison, much of the variability between the two instruments and algorithms occurs during events inhibit a reliable estimation (e.g., fog, precipitation) of MLH (as seen in Fig. 13).

The most commonly used statistical techniques used for comparing two datasets depended on two key assumptions: data

were normally distributed and homoscedastic. The CL51 and MPL MLH 5-min averaged datasets were confirmed to be non-normal via the Kolmogorov-Smirnov test and passed Levene's test for homoscedasticity (p-value 0.39). Therefore, similarity between the two corresponding probability distributions was determined using the two-sample Kolmogorov-Smirnov test. It was determined that the 5-min averaged MPL and CL51 datasets were statistically different (p $\ll$ 0.01), regardless of filtering and averaging. However, when considering 1-hour averaged data that were filtered to remove data with large relative standard

deviations ($\geq 0.20$) and MLH $\leq 0.5$ km, the two datasets were statistically indistinguishable (p 0.8). While we cannot account for the bias induced by these low-altitude MLH values it is quite clear that they significantly influence the intercomparison. Given that this is the first intercomparison of these two instruments and algorithms, it is not surprising that a significant difference was identified in this regime.



**Figure 10.** Potential temperature and Cl51 backscatter profiles collected at the BAO-Tower site. Horizontal lines indicate MLH as determined by BLView.





**Figure 11.** Potential temperature and Cl51 backscatter profiles collected at the Golden NREL site. Horizontal lines indicate MLH as determined by BLView.





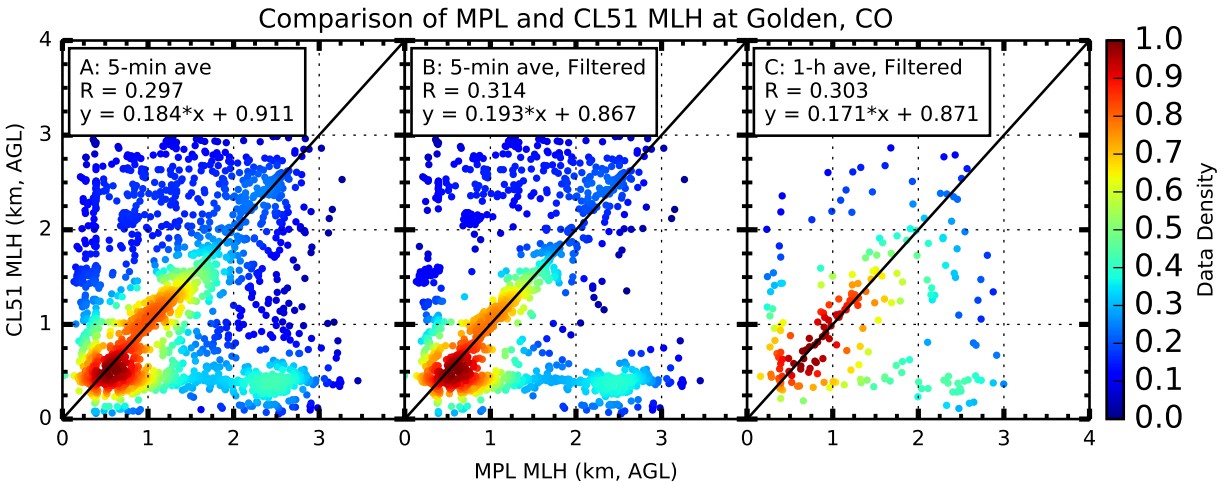

**Figure 12.** Correlation and data-density plots for the CL51 and MPL MLH estimates from Golden, Colorado.

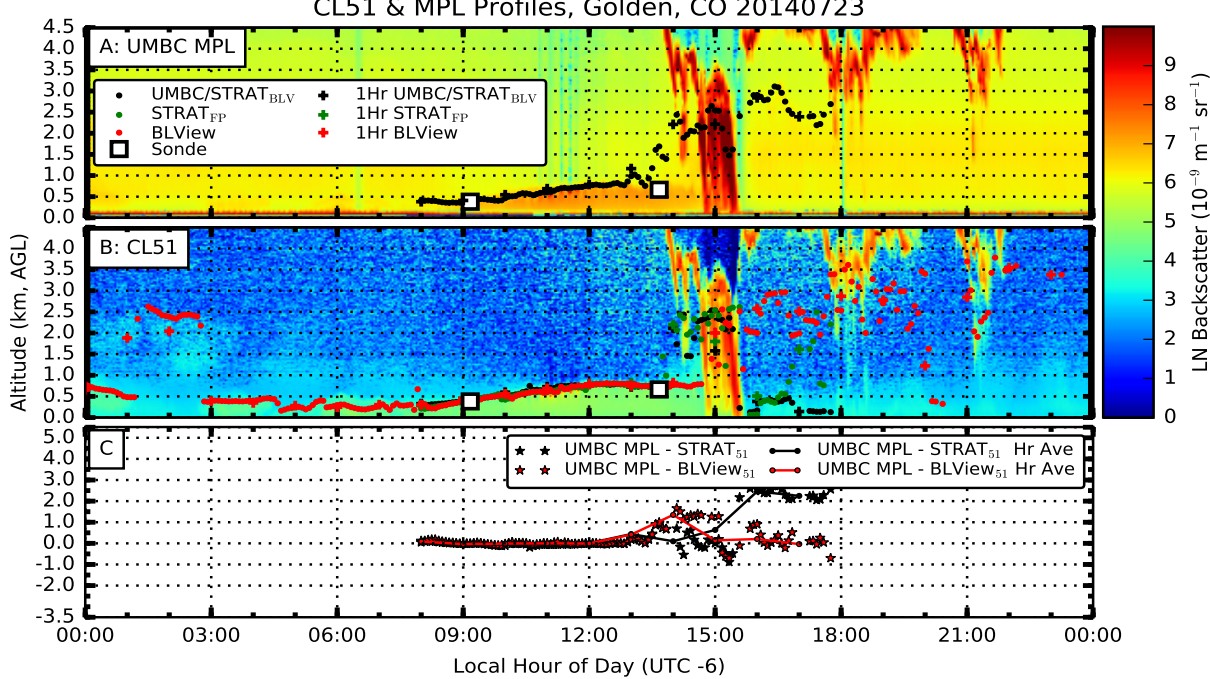

**Figure 13.** Comparison of the CL51 and UMBC MPL profiles for 23/27-July 2014 at the Golden, Colorado site.



## 6   Conclusions

A CL51-focused intercomparison of different ABL/MLH methodologies was performed at three different sites that experience different meteorological, aerosol, and emission conditions. The CL51 MLH results were compared with ABL from radiosondes at all three locations; as well as an MPL at the Golden, Colorado site.

Two collection methods and processing algorithms were tested for the CL51 MLH calculation. We demonstrated that the data-collection method played an insignificant role in MLH estimation when the datasets were processed using a common algorithm. Furthermore, the choice of processing algorithm played a significant role in MLH estimation. Therefore, we recommend that, for ceilometer and LIDAR networks, a common MLH processing algorithm be employed. Agreement between the different algorithm products might be dictated, to a large degree, by local atmospheric fluctuations and interferential events (e.g. fog), which should be a topic for future investigation.

A total of 53 potential temperature profiles from radiosondes were used to evaluate the CL51. While the 53 radiosondes were spread across 3 sites, this represents a robust data set of soundings. Overall, the met-sonde-derived ABL was higher than the CL51 MLH. Comparison of MLH from the CL51 versus met-sondes shows the CL51 performed best at the CAPABLE research site (non-filtered R = 0.81, filtered R = 0.82), a moderately polluted coastal site primarily influenced by a combination of sulfate and marine aerosols. Both the Golden and BAO-Tower sites are located in cleaner environments than CAPABLE and show good correlation between the CL51 and met-sondes (Golden filtered R = 0.74, BOA non-filtered R = 0.63, filtered R = 0.38) with two early morning sondes at the Golden site strongly influencing the non-filtered correlation (R = -0.28). These two sondes measured a very shallow boundary layer, < 500 m, while the CL51 identified the MLH above 2 km, which was likely due to residual aerosol layers aloft. The lower correlations at the Colorado sites (Golden and BAO-Tower) were likely due to the sites proximity to the Rocky Mountains. Complex atmospheric flow patterns, which are driven by the Rocky Mountains to the west of the Front Range area, can induce the formation of distinctive dynamic features such as up and downslope flows (Bossert et al., 1989; Bossert and Cotton, 1994; Tripoli and Cotton, 1989). The Golden site likely experienced greater up- and down-slope flows than the BOA-Tower site because of its location along the slope of the mountains and on a mesa. Such local orographic influences likely impacted or challenged the well-mixed assumption required to compare thermodynamic ABL measured via potential temperature and MLH measured via aerosol backscatter in the current study. These influences should be made a consideration in future intercomparisons.

The results of the CL51 versus the UMBC algorithm that was run on MPL data showed low correlation (R = 0.3).However, the majority of coincident MLH observations from both instruments were clustered around the 1:1 line in the regression plots. When data-filtering criteria were applied, the two data sets were statistically indistinguishable (p > 0.8). Additional analysis is planned to further explore the cause of the low correlation. However, the MLH from the CL51 and MPL agree well when there is a well-defined MLH.



*Acknowledgements.* Funding for this work was provided by the NASA Applied Sciences Program, EPA collaborations under an EPA-LaRC memorandum of agreement, GEO-CAPE mission studies, and Langley Innovative Partnership Program. T. N. Knepp was funded through the STARS-III contract. Although this paper has been reviewed by the EPA and approved for publication, it does not necessarily reflect EPA policies or views. Mention of trade names or commercial products does not constitute endorsement or recommendation for use.





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
