# Peer review of "Assessment of Mixed-Layer Height Estimation from Single-wavelength Ceilometer Profiles"

_Atmospheric Measurement Techniques, 2017_

## Referee Comment (RC1) · Anonymous Referee #1 · 21 Jun 2017

Review of "Assessment of Mixed-Layer Height Estimation from Single-wavelength Ceilometer Profiles" by Knepp et al. (2017) in Atmospheric Measurement Techniques Discussions:

Summary:

This paper presents an intercomparison study of Vaisala CL51 ceilometer-derived mixed layer heights (MLH) using different data collection and processing procedures, along with comparisons against co-located radiosonde and Micropulse Lidar measurements. CL51 data were collected at three locations (two in Colorado, one at LaRC in Hampton, VA) using both the proprietary Vaisala software BLView, and a custom

[Figure]

Python logging script. The authors find that while there is little MLH dependence on choice of the two data collection techniques, significant discrepancies in calculated MLH heights are observed when comparing two different data processing techniques. As with the actual data collection, one processing method is from the proprietary Vaisala BLView software, and the other is open-source (STRAT) and currently used in European networks.

Comparisons of the two CL51 data collection methods processed only with STRAT show good agreement (MLH r = 0.72-0.87). However, correlation coefficients drop to r $\sim$ 0.5 and lower when the CL51 data are processed with the different STRAT and BLView processing algorithms. Comparisons between radiosonde-derived atmospheric boundary layer (ABL) heights and CL51 MLH were decent, with exceptions during morning hours. CL51 and MPL comparisons at the Golden, CO, seemed to be a bit worse than the radiosonde comparisons.

Given the dependence of calculated MLH on the data processing algorithm, the authors recommend a single processing algorithm be used for ceilometer networks. The authors seem to imply, but do not state, that open-source logging of data is preferred so the full 15.4 km CL51 backscatter profile (rather than only up to 4.5 km) can be collected.

General Comments and Recommendation:

Though the reader will be left wondering which data collection and processing methods are best in terms of validation with independent data, this is an important and well-motivated study that will lead to those types of efforts. Agreement and consistency in data collection and processing is something with which many measurement networks struggle. The BLView software appears to be a bit of a black box, and elucidating differences between it and open-source methods are a priority. Given upcoming efforts to include MLH measurements as part of the US air quality monitoring network, this type of technical analysis is a necessary step.

I have several comments which I think should be mostly minor. These involve being as clear as possible such as when describing averaging time and vertical resolutions, and exactly which CL51 data processing algorithm is being depicted in each figure and why (e.g. Figures 9 and 12). After these are addressed I feel comfortable recommending publication in AMT.

Specific/Technical Line-by-Line Comments:

Page 2, Line 21-23: Do you have a reference for this statement from the US NRC?

Page 2, Line 24: This reference seems incomplete.

Page 2, Lines 29-34: These lines are written in passive voice. Please rewrite.

Page 2, Line 34: The way this line is written makes it seem like you are comparing three CL51s from Colorado against sondes from CAPABLE.

Page 3, Line 26: It's never stated why BLView truncates data at 4.5 km. Are there concerns about measurement uncertainties or S/N ratios at higher altitudes? I realize this probably doesn't have an effect on the MLH calculations.

Page 4, Line 12: Are the 1 min and 30 m resolutions from the MPL what you've chosen to record specifically for this study? Please state.

Page 5, Line 25: Delete "to be."

Page 5, Line 26: clouds not cloud.

Page 6, Figure 2: The .5s are missing on the y-axis labels.

Page 7, Line 3: What parameters specifically? Averaging time? Vertical resolution?

Page 7, Line 10: "A detailed description of the UMBC algorithm has been published in Compton et al. (2013)."

Page 7, Lines 10-14: These lines contain jargon that receives no other mention. You can probably tack the single sentence on line 10 to the end of the previous paragraph

and delete the rest.

Page 8, Line 2: farther not further.

Page 9, Line 11-13: Are you saying that there were two CL51s at the BAO-Tower? I'm confused about the instrument set up here.

Page 9, Line 15: "CL51 data were collected..."

Page 9, Line 19: A figure showing average diurnal MLH from each of the three sites would be very helpful here and would give context for the statement that Golden often does not observe a well-developed boundary layer.

Page 9, Line 25: Only the CL51 and MPL data were averaged to 5 min resolution, correct? There are a lot of time and vertical resolution averaging numbers being thrown around and they should all be clear.

Page 10, Line 23: "...the standard deviation of MLH was calculated..."

Page 11, Figure 3: Somewhere in the text it would be useful to state that all times presented are in local standard time.

Page 11, Lines 4-9, Figure 5: I found Figure 5 to be confusing and in need of some clarification. How should this figure be interpreted? That variability within the 5 min measurement period is generally very low when the methods agree, and peaks when the difference between the two methods is between .5 and 1km? Shouldn't relative standard deviation ($\sigma$/xbar) be unitless? It has units of km on Figure 5. Please clarify.

Page 15, Figure 7: The color bar and what's plotted on the z-axis are not the same as Figure 6. Did you mean to plot data density rather than relative standard deviation? The current Figure 7 seems to present similar data as Figure 5 in a different way.

Page 19, Line 8: According to Figure 9, the correlations are actually 0.81, and 0.82, not 0.82 and 0.83.

[Figure]

Page 19, Figure 9: Do these statistics significantly change based on processing method? What do the error bars represent? In general, many of the figures would benefit from more detailed captions.

Page 19, Figure 9: Can you add additional plots to Figure 9 showing the STRAT and sonde comparison?

Page 19, Figure 9: Please adjust the axes to less than 7 km so spread in the data can be better visualized.

Page 19, Figure 9: I'm curious what the correlation of MLH with all sondes is. Better or worse than the individual sites?

Page 19, Line 20: "It is somewhat surprising that the filtered..."

Page 19, Line 20: It's difficult to definitively say that correlations at one site are "better" than another given the small sample size. What are the 90 or 95% confidence interval limits on these correlations?

Page 20, Line 20: Yes, there is similar behavior at CAPABLE in the comparisons on Figure 8. This is worth future exploration for the BLView output. Did you look at STRAT processing vs. the MPL? Does this invariance feature disappear? Can you add additional plots to Figure 12 showing the MPL vs. STRAT?

Page 20, Line 21: "Removal of MLH below 500 m..."

Page 21, Figure 10: Why do the CL51 profiles only go up to 3 km here? Same with Figure 11.

Page 23, Figure 13: Please adjust the y-axis on plot C so we can better observe the variability in MLH differences.

Page 24, Line 17: Should be 0.58 not 0.38.

Page 24, Line 20: sites' not sites

Page 24, Line 22: A good up-to-date reference from DISCOVER-AQ Colorado on these types of circulations and how they affect pollution distribution is Sullivan et al. (2016, JGR): http://onlinelibrary.wiley.com/doi/10.1002/2016JD025229/full

---

## Referee Comment (RC2) · Anonymous Referee #2 · 23 Jun 2017

The authors compared different (proprietary and open-source) data collection and assessment algorithms for mixing-layer height (MLH) from optical remote sensing with ceilometers. Partly, the results have been compared to radiosonde ascent data and systematic differences have been found. The main result is that a common algorithm should be used in a larger ceilometer network for assessing MLH.

The study is meaningful and will help to establish well-organised ceilometer networks. Thus, it deserves publication. But a few points should be addressed in more detail or depth before a final publication of this manuscript can be recommended.

The main point is that an indirect MLH assessment method (the one using aerosol

backscatter data from ceilometers) is compared to a direct method (the evaluation of profile data from radiosonde ascents). Furthermore, ceilometers are remote sensing devices while radiosondes are in-situ probes. These two important differences must produce different results apart from the additional differences in the data collecting methods and the used evaluation algorithms. This must be discussed (along the three points mentioned below) and made transparent to the readers.

(1) Aerosols are used as a tracer for the vertical structure of the atmospheric boundary layer when evaluating MLH from aerosol backscatter intensities. It should be kept in mind that atmospheric particles need some time to adapt to a changing vertical structure of the atmospheric boundary layer (see, e.g., the lower right frame in Fig. 1 in Emeis and Schäfer 2006). Therefore, it might be advisable to compare radiosonde results to ceilometer results obtained in the hour after (or even in the two hours after) the radiosonde ascent.

(2) Horizontal advection of atmospheric particles can deteriorate the relation between the vertical structure of the boundary layer and the vertical profile of aerosol backscatter intensity.

(3) Radiosonde data usually have some sort of a hysteresis. The sensors need some time to adapt to the environmental conditions during the ascent. This could lead to a small bias towards higher MLH.

A minor point is that the Spanish word "mesa" should be explained to readers not acquainted to the topography of the surroundings of Boulder, Colorado.

Reference

Emeis, S. and K. Schäfer, 2006: Remote sensing methods to investigate boundary-layer structures relevant to air pollution in cities. Bound-Lay. Meteorol., 121, 377-385.

---

## Author Comment (AC1) · 3 Aug 2017

[amtd]copernicus

We appreciate the thorough review from referee #1. The manuscript has been updated to implement these recommendations as described below.

1. "... being as clear as possible such as when describing averaging time and vertical resolutions, and exactly which CL51 data processing algorithm is being depicted in each figure..."

[Figure]

(a) Clarification was made throughout the text to aid the reader in knowing what averaging, resolutions, and algorithms are being used.

2. Page 2, Line 21-23: Do you have a reference for this statement from the US NRC?

    (a) Reference has been included.

3. Page 2, Line 24: This reference seems incomplete

    (a) Reference corrected

4. Page 2, Lines 29-34: These lines are written in passive voice. Please rewrite

    (a) Paragraph rewritten

5. Page 2, Line 34: The way this line is written makes it seem like you are comparing three CL51s from Colorado against sondes from CAPABLE.

    (a) Sentence corrected.

6. Page 3, Line 26: It's never stated why BLView truncates data at 4.5 km. Are there concerns about measurement uncertainties or S/N ratios at higher altitudes? I realize this probably doesn't have an effect on the MLH calculations.

    (a) That is unknown to us and is one of the challenges in using proprietary software as it remains a black box. Any comment we provide on this would be speculative, so we will not comment.

7. Page 4, Line 12: Are the 1 min and 30 m resolutions from the MPL what you've chosen to record specifically for this study? Please state

    (a) Stated

8. Page 5, Line 25: Delete "to be."

   (a) Deleted

9. Page 5, Line 26: clouds not cloud.

   (a) Changed

10. Page 6, Figure 2: The .5s are missing on the y-axis labels

    (a) y-tick labels corrected

11. Page 7, Line 3: What parameters specifically? Averaging time? Vertical resolution?

    (a) Correct. Now specified in text.

12. Page 7, Line 10: "A detailed description of the UMBC algorithm has been published in Compton et al. (2013)."

    (a) Recommendation implemented.

13. Page 7, Lines 10-14: These lines contain jargon that receives no other mention. You can probably tack the single sentence on line 10 to the end of the previous paragraph and delete the rest.

    (a) Change implemented.

14. Page 8, Line 2: farther not further

    (a) Change implemented.

15. Page 9, Line 11-13: Are you saying that there were two CL51s at the BAO-Tower? I'm confused about the instrument set up here.

(a) No, they housed the CL51 used in the current study. Text changed to "the CL51" instead of "a CL51" to indicate this.

16. Page 9, Line 15: "CL51 data were collected ..."

    (a) Change implemented.

17. A figure showing average diurnal MLH from each of the three sites would be very helpful here and would give context for the statement that Golden often does not observe a well-developed boundary layer.

    (a) New figure (Fig. 3) inserted and text added within body.

18. Page 9, Line 25: Only the CL51 and MPL data were averaged to 5 min resolution, correct? There are a lot of time and vertical resolution averaging numbers being thrown around and they should all be clear.

    (a) That is correct. Due to the nature of sonde data we cannot resample to a longer time period. Clarification is made within the text.

19. Page 10, Line 23: "... the standard deviation of MLH was calculated ..."

    (a) Recommended change implemented

20. Page 11, Figure 3: Somewhere in the text it would be useful to state that all times presented are in local standard time

    (a) Statement added in analysis section

21. Page 11, Lines 4-9, Figure 5: I found Figure 5 to be confusing and in need of some clarification. How should this figure be interpreted? That variability within the 5 min measurement period is generally very low when the methods agree, and peaks when the difference between the two methods is between .5 and 1km?

Shouldn't relative standard deviation ( $\sigma$ /xbar) be unitless? It has units of km on Figure 5. Please clarify.

(a) You are correct, $\sigma$ / xbar should be unitless. While this figure is interesting, it is only mentioned in the text once and we do not feel that it adds significantly to the manuscript. Rather, inclusion only distracts the reader and may cause unnecessary confusion. The intention of including this figure was to further support the selection of filter criteria, though we feel these criteria are adequately supported without this figure. The figure was removed from the manuscript.

22. Page 15, Figure 7: The color bar and what's plotted on the z-axis are not the same as Figure 6. Did you mean to plot data density rather than relative standard deviation? The current Figure 7 seems to present similar data as Figure 5 in a different way.

(a) We appreciate the reviewer's sharp eye to catch this. Figure 7 was properly labeled, but the caption needed updated and supporting text within the manuscript's body needed clarified. The caption was updated and descriptive text was added within the paragraph beginning with "For regulatory and modeling applications...".

23. age 19, Line 8: According to Figure 9, the correlations are actually 0.81, and 0.82, not 0.82 and 0.83.

(a) Statistics corrected.

24. Page 19, Figure 9: Do these statistics significantly change based on processing method? What do the error bars represent? In general, many of the figures would benefit from more detailed captions.

Page 19, Figure 9: Can you add additional plots to Figure 9 showing the STRAT and sonde comparison?

(a) Yes and no. Text was added to explain this, as were two additional figures. The STRAT algorithm gets tricked in places (as discussed in the added text) and will need further refining before it is capable of operating to lesser degrees of human intervention. However, this may be a strength of the open source software paradigm in that the end user can adjust the algorithm to train it for specific purposes, if desired.

25. Page 19, Figure 9: Please adjust the axes to less than 7 km so spread in the data can be better visualized.

    (a) Figure changed.

26. Page 19, Figure 9: I'm curious what the correlation of MLH with all sondes is. Better or worse than the individual sites?

    (a) That is an interesting thought. The composite correlations are not much different than the weighted average of the individual statistics. A "composite" dataset has been added to Table 5.

27. Page 19, Line 20: "It is somewhat surprising that the filtered..."

    (a) Change implemented

28. Page 19, Line 20: It's difficult to definitively say that correlations at one site are "better" than another given the small sample size. What are the 90 or 95% confidence interval limits on these correlations?

    (a) You are correct that marking one set as "better" is challenging due to the small sample sizes. However, calculation of a Pearson's coefficient of correlation confidence interval is highly unreliable due to the size of the data sets. We do not feel this would be representative of the true population statistics, so we will forbear including this statistic here. This may be a beneficial statistic to include in future work that involves larger data sets.

29. Page 20, Line 20: Yes, there is similar behavior at CAPABLE in the comparisons on Figure 8. This is worth future exploration for the BLView output. Did you look at STRAT processing vs. the MPL? Does this invariance feature disappear? Can you add additional plots to Figure 12 showing the MPL vs. STRAT?

    (a) Similar behavior is seen with the STRAT and BLView algorithms. An additional figure has been added to show CL51 comparison with MPL via the two algorithms.

30. Page 20, Line 21: "Removal of MLH below 500 m..."

    (a) Suggested correction implemented.

31. Page 21, Figure 10: Why do the CL51 profiles only go up to 3 km here? Same with Figure 11.

    (a) The focus of the manuscript is on the mixed layer or boundary layer, which is well below 3 km throughout the study. As nothing of relevance is within the 3 km+ profile the profile was truncated to prevent the figure from becoming overly crowded and allow inclusion of text within the upper-left corners of each figure.

32. Page 23, Figure 13: Please adjust the y-axis on plot C so we can better observe the variability in MLH differences

    (a) Axis changed to show full-scale variability.

33. Page 23, Figure 13: Please adjust the y-axis on plot C so we can better observe the variability in MLH differences.

    (a) Change implemented.

34. Page 24, Line 20: sites' not sites

(a)

35. Page 24, Line 22: A good up-to-date reference from DISCOVER-AQ Colorado on these types of circulations and how they affect pollution distribution is Sullivan et al. (2016, JGR...

    (a) Reference included

---

## Author Comment (AC2) · 3 Aug 2017

[amtd]copernicus

We appreciate the reviewer's comments, suggestions, and taking the time to review the manuscript. We address the comments below.

1. Aerosols are used as a tracer for the vertical structure of the atmospheric boundary layer when evaluating MLH from aerosol backscatter intensities. It should be kept in mind that atmospheric particles need some time to adapt to a changing vertical structure of the atmospheric boundary layer (see, e.g., the lower right

[Figure]

frame in Fig. 1 in Emeis and Schäfer 2006). Therefore, it might be advisable to compare radiosonde results to ceilometer results obtained in the hour after (or even in the two hours after) the radiosonde ascent.

(a) We agree that time is required for particle distribution to adapt to changing atmospheric thermodynamics. However, these changes will be most notice-able during transition times (e.g. dawn and dusk). The number of data points from our dataset during these transition times is too sparse to generate a statistically meaningful analysis. The bulk of our data were collected outside transition events, when the MLH/ABL is comparatively stable. Therefore we consider the analysis, as presented, to be correct and would implement the reviewer's suggestion for data collected during transition events.

2. Horizontal advection of atmospheric particles can deteriorate the relation between the vertical structure of the boundary layer and the vertical profile of aerosol backscatter intensity.

(a) Now addressed.

3. Radiosonde data usually have some sort of a hysteresis. The sensors need some time to adapt to the environmental conditions during the ascent. This could lead to a small bias towards higher MLH.

(a) Now addressed

4. A minor point is that the Spanish word "mesa" should be explained to readers not acquainted to the topography of the surroundings of Boulder, Colorado.

(a) Mesa is the proper term for a geographic structure. A very brief description was added to the text.

---

## Author Response (AR1)

**Assessment of Mixed-Layer Height Estimation from Single-wavelength Ceilometer Profiles**

T. Travis N. Knepp1,2, James J. Szykman3,4, Russell Long3, Rachelle M. Duvall3, Jonathan Krug3, Melinda Beaver3, Kevin Cavender3, Keith Kronmiller5, Michael Wheeler5, Ruben Delgado6, Raymond Hoff6, Timothy Berkoff2, Erik Olson7, Richard Clark8, Daniel Wolfe9, David Van Gilst10, and Doreen Neil2

1Science Systems and Applications Inc., Hampton, Virginia 23666, USA
2NASA Langley Research Center, Hampton, Virginia 23681, USA
3US EPA, Research Triangle Park, Durham, North Carolina 27709, USA
4Currently assigned to NASA Langley Research Center, Hampton, Virginia 23681, USA
5Jacobs Technology Inc., Tullahoma, Tennessee 37388, USA
6Joint Center for Earth Systems Technology, University of Maryland Baltimore County, Baltimore, Maryland 21250, USA
7Space Science and Engineering Center, University of Wisconsin-Madison, Madison, Wisconsin 53706, USA
8Department of Earth Sciences, Millersville University, Millersville, Pennsylvania 17551, USA
9NOAA/ESRL Physical Sciences Division, Boulder, Colorado 80305, USA

*Correspondence to:* T. N. Knepp (travis.n.knepp@nasa.gov)

Abstract. An assessment of differing Differing boundary/mixed-layer height measurement methods was performed were assessed in moderately-polluted and clean environments, with a focus on the Vaisala CL51 ceilometer. This intercomparison was performed as part of ongoing measurements at the Chemistry And Physics of the Atmospheric Boundary Layer Experiment (CAPABLE) site in Hampton, VA\_Virginia and during the 2014 Deriving Information on Surface Conditions from

- 5 Column and Vertically Resolved Observations Relevant to Air Quality (DISCOVER-AQ) field campaign that took place in the Denver, CO area and around Denver, Colorado. We analyzed CL51 data that was were collected via two different methods (i.e. via the BLView software, which applied correction factors, and simple terminal emulation logging) to determine the impact of data collection methodology. Further, we evaluated the STRAT STRucture of the ATmosphere (STRAT) algorithm as an open-source alternative to BLView (NOTE: note that the current work presents an evaluation of the BLView and STRAT
- 10 algorithms and does not intend to act as a validation of either). A common filtering criteria was Filtering criteria were defined according to the AMLHchange in mixed-layer height (MLH) distributions for each instrument and algorithm and were applied throughout the analysis to remove high-frequency fluctuations from the MLH retrievals, and was applied throughout the analysis. Of primary interest was determining how the different data-collection methodologies and algorithms compare to each other and to radiosonde-derived boundary-layer heights when deployed as part of a larger instrument network. We determine
- 15 that data collection\_determined that data-collection methodology is not as important as the processing algorithm , and that much of the algorithm differences may might be driven by impacts of local meteorology and precipitation events that pose algorithm difficulties. The results of this study show that for LIDAR-based a common processing algorithm is necessary for Light

Detection And Ranging (LIDAR)-based MLH intercomparisons, and for ceilometer-network operation , a common processing algorithm is necessary and that sonde-derived boundary layer heights are higher (10–15% at mid-day) than LIDAR-derived mixed-layer heights. We show that averaging the retrieved MLH to one-hour resolution (i.e. as necessary-1-hour resolution (an appropriate time scale for a priori data for-model initialization) significantly improved correlation between differing instruments and differing algorithms.

5

**1 Introduction**

The atmospheric boundary layer (ABL) is the lowermost portion of the troposphere that is directly influenced by the Earth's surface and responds to surface forcing of heat, moisture, pollutant emissions, and momentum on timescale of an a timescale of 1 hour or less (?). The ABL can be defined via by a number of criteria depending on the particular interest (e.g. the thermo-

- 10 dynamic boundary layer, chemical boundary layer (CBL), aerosol mixed layer, etc.). Traditionally, the ABL has been ). The ABL is typically defined by thermodynamic data (i.e., potential temperature and/or skew-T plot) obtained from meteorological sondes. While meteorological sondes have excellent vertical resolution, the temporal resolution is generally poor, and ongoing regular sonde launches are labor intensive, and coverage is limited. Conversely, mixed-layer heights (MLH), as calculated from backscatter LIDAR instruments Light Detection And Ranging (LIDAR) instruments, provide both excellent vertical and
- 15 temporal resolution. Typical analysis of LIDAR data involves identification of gradients within the aerosol profile (?), which is generally considered to be a marker for the MLH. With respect to air quality, the top of the ABL often acts like a lid within on the lowest layer of the atmosphere and temporarily traps the majority of near-surface anthropogenic and biogenic emissions. As a result, the vertical distribution of ambient air pollutants, and associated precursors, within the ABL and lower-troposphere are strongly influenced by the height of, and vertical mixing within, the ABL.
- 20 ABL variability complicates quantitative determination of surface trace-gas levels from a remote-sensing platform (???????). Therefore, properly accounting for ABL variability from a continuous measurement system such as LIDAR will provide invaluable information to policy, health, modeling, and remote-sensing communities for applications sensitive to the vertical profiles of tracers (???). In 2009, the United States National Research Council highlighted planetary (atmospheric) boundary layer\_ABL height as a high priority observation needed to address current inadequacies at the improve meso-scale for improved
- 25 predictions of air quality, short-range severe-weather forecasting, and regional climate modeling (?). More recently, The the National Plan for Civil Earth Observation (?) called out the need called for improved observation density and sampling of the boundary layer (?). In 2015, as part of the revisions to the ozone (O3) National Ambient Air Quality Standard (NAAQS), EPA Standards, the U.S. Environmental Protection Agency (EPA) finalized a new requirement under the Photochemical Assessment Monitoring Stations (PAMS) program for the collection of continuous mixing layer height MLH observations. By 2019, the
- 30 PAMS program will involve the implementation of approximately fifty air-quality sites around in the United States , providing measurements of MLH on a continuous basis that provide continuous MLH.

From 2011 through 2014 NASA conducted the Deriving Information on Surface Conditions from Column and Vertically Resolved Observations Relevant to Air Quality (DISCOVER-AQ) Earth Venture Suborbital Mission with four field deployments: Baltimore/Washington region of Maryland during 2011; the San Joaquin Valley (SJV) of California during January-February 2013; Houston, Texas during September 2013; and the Front Range region of Colorado in July-August 2014. A primary objective of DISCOVER-AQ was to investigate the use of satellite remote sensing ability to inform air quality at the surface.

- 5 Since the ABL limits vertical exchange of primary pollutants, and controls near-surface pollutant concentrations, the ABL height can directly influence air quality and chemistry. Therefore, measurements during these missions focused on the vertical distribution of trace gases and aerosols within the ABL and lower troposphere, and the diurnal variability of these distributions in conjunction with the ABL.
- ABL variability poses a complication in quantitative determination of surface trace-gas levels from a remote-sensing platform 10 (??????). Therefore, properly accounting for ABL variability from a continuous measurement system such as Light Detection And Ranging (LIDAR) will provide invaluable information to policy, health, modeling, and remote-sensing communities for applications sensitive to the vertical profiles of tracers (???). Herein is presented results from ? showed that intercomparison of ceilometer data is not a straight-forward endeavor. An intercomparison of ceilometer instrumentation was carried out in support of upcoming PAMS monitoring requirements. Results from an intercomparison of three backscatter LIDAR 's instruments from
- 15 the 2014 DISCOVER-AQ field campaign in Colorado (low aerosol load) and <del>coincident sonde launches from</del> the Chemistry and Physics of the Atmospheric Boundary Layer Experiment (CAPABLE) site at NASA's Langley Research Center (LaRC, moderate aerosol load) in Hampton, VA Virginia are presented herein.

**2 Instrumentation**

**2.1 CL51**

- The Vaisala (Vantaa, Finland) CL51 ceilometer is a single-wavelength (eye safe Class 1M InGaAs diode LASER-laser emitting at 910  $\pm$  10 nm, pulsed at 6.5 kHz with a 110 ns pulse width <del>, and with</del> average pulse power of 19.5 mW, with and an avalanche photodiode detector centered at 915 nm), single-lens, LIDAR system originally designed to report cloud-base heights and visibility. More recently, ceilometers have been used to estimate MLH (?????????). These ceilometers have 10 m vertical resolution (with 10 m overlap) to a maximum altitude of 15.4 km ( $\pm$  greater of 1% or 5 m precision)<del>,</del> all altitudes are with
- 25 respect to ground level) and up to 2 s temporal resolution (depending on the control software), though profiles are generally averaged over 16–36 s to improve the signal-to-noise (see Sec. ratio (see section 3.1 for more details). An example backscatter plot that includes increased signal at 3 km due to transport of smoke from a Canadian forest fire is presented in Fig. 1.

The CL51 was designed to operate continuously, regardless of meteorological conditions, in an autonomous manner with minimal user support. Due to the emission wavelength's proximity to the near-infrared water vapor bandsthese ceilometers.

30 ceilometers operating at the stated wavelengths experience water vapor interference, thereby mitigating lessening their utility in retrieval of aerosol optical properties. However, the interference on aerosol profile and MLH estimation is negligible (?).

Two CL51's CL51s were deployed as part of the 2014 DISCOVER-AQ mission in Colorado (Golden, and Erie, COColorado). Before and after deployment, these ceilometers were stationed at CAPABLE set up to continually collect data at the CAPABLE site and the EPA Ambient air Air Innovative Research Site (AIRS) in Durham, NC, continually collecting dataNorth Carolina. The ceilometers were collocated with meteorological sonde (met-sonde) launch sites during the DISCOVER-AQ campaign and at CAPABLE the CAPABLE site, allowing a direct intercomparison of the sonde and LIDAR ABL/MLH methodologies.

5 Furthermore, during the DISCOVER-AQ campaign the ceilometers were collocated with other LIDAR instruments. Intercomparisons are presented belowin section 5.

**2.1.1 Ceilometer Full-profile Collection**

The BLView software not only provides Vaisala standard MLH retrieval is based on a proprietary wavelet/gradient technique built into the logging/analysis software BLView. The BLView software provides not only logging and data analysis (e.g. MLH

- 10 and cloud-height estimates) <del>, but also provides data-logging/but also</del> archiving capability. While the CL51 reports backscatter up to 15.4 km, BLView truncates the profile data collection\_data-collection\_at 4.5 km. Generally speaking, there is little need to collect higher-altitude backscatter data for reprocessing purposes due to the relative simplicity of detecting cloud bases. However, failure to log the full-profile reduces the precluding ability to monitor upper-troposphere/lower-stratosphere (UTLS) transport of aerosol, smoke, or ash from major events. Therefore, a full-profile collection method that can run side-by-side with
- 15 the standard data-collection software was developed and implemented.

Data transmission from the ceilometer to the logging computer was achieved over a simple by splitting an RS-232 connection that can be split into two ports on the logging computer; one port logging to BLView , and the other logging to a custom script (e.g. as written in Python, or terminal emulation). The primary drawback of using a secondary script to log the full profile (as opposed to logging in BLView) is the inability to apply ealibration coefficients proprietary calibration coefficients that are

20 built into the BLView software to the logged data. However, as shown in subsequent sections, this impacts neither the MLH estimates nor the general profile shape substantially.

**2.2 MPLMicropulse LIDAR**

Elastic LIDAR observations were performed using a Sigma Space (Lanham, Maryland) Micropulse LIDAR (MPL), previously described in by ? and ?. Briefly, the MPL transmitter consists of an eye-safe Nd:YLF laser emitting at 527 nm, and pulsed at 2.5

- kHz and an average with a pulse power of  $6 10 \mu J_{\tau}$ . It has a software programmable vertical resolution, with possible values of 15, 30, and 75 m (up to 25 km), and temporal resolutions ranging from 1 s – 15 minutesto 15-min. The receiver consists of a 178 mm telescope that collects the backscattered light, which is then focused onto a photon counting silicon avalanche photodiode (APD). The APD output is recorded by a field programmable gate array (FPGA) data system that enables display and storage of range dependent averaged average count rates on a laptop computer. The raw data are converted to aerosol attenuated
- 30 backscatterby taking into account instrumental factors that include corrections for, correcting for instrumental factors such as detector dead time, geometrical overlap, background subtraction, and range-squared normalization. Recorded LIDAR profiles have temporal and vertical resolution of one minute 1 min and 30 meters, respectively. The m, respectively, as set by the UMBC team for the DISCOVER-AQ campaign. MPL is used for continuous recording of aerosol profiles , and optical properties, and calculating MLH values.